# Climatology, long-term variability and trend of resolved gravity wave drag in the stratosphere revealed by ERA 5

Zuzana Procházková[1], Radek Zajíček[1], and Petr Šácha[1]

[1]Department of Atmospheric Physics, Faculty of Mathematics and Physics, Charles University, Prague, Czech Republic

**Correspondence:** Zuzana Procházková (prochazkova@karlin.mff.cuni.cz) and Petr Šácha (petr.sacha@matfyz.cuni.cz)

**Abstract.** Internal gravity waves have a well-known importance for atmospheric dynamics, transport and coupling between atmospheric layers and their parameterized forcing affects the circulation in climate models, especially in the stratosphere. The statistical features, spatial distribution, short- and long-term variability of the parameterized gravity wave drag were studied extensively. Yet, little is known about the gravity wave drag in the real atmosphere. Challenges arise when attempting to constrain gravity wave drag using observational data, leading to the widespread use of wave activity proxies. Moreover, our limited observational capabilities hinder comprehensive assessments of global, long-term changes in stratospheric dynamical quantities.

This study presents a quasi-observational analysis of resolved gravity wave drag climatology, variability, and trends in the stratosphere. We employ a state-of-the-art methodology for gravity wave drag estimation, applying it to ERA 5, a latest-generation atmospheric reanalysis that resolves a substantial portion of the gravity wave spectrum (wavelengths from a few hundred to a few thousand kilometers). The results are provided in the traditional zonal mean perspective and, for the first time in the literature, we focus also on regional drag estimates over major orographic hotspots taking fully into account the drag from lateral gravity wave propagation. Overall, our study represents a first step towards validating climatology and variability of parameterized gravity wave drag in climate models.

## 1 Introduction

Internal gravity waves (GWs) are an ubiquitous, multi-scale phenomenon in the atmosphere with a pronounced influence on atmospheric dynamics, transport and vertical coupling in general (Fritts and Alexander, 2003; Achatz, 2022). Because GWs exist at and influence processes across a wide range of scales, they pose a challenge for numerical global atmospheric models. A significant portion of the GW spectrum often has wavelengths smaller than the model's effective resolution. Hence, momentum deposition and other possible effects of the unresolved part of the spectrum have to be parameterized (Achatz et al., 2024). GW parameterization schemes rely on various simplifications that are well justified at leading order, but also employ several tuneable parameters that are poorly constrained by observations (Plougonven et al., 2020).

Our understanding of the pronounced effects of parameterized gravity wave drag (GWD) in Earth system models has been significantly enhanced by recent research. For non-orographic GW parameterizations, for instance, Richter et al. (2020) have shown that the projected Quasi-biennial oscillation (QBO) changes are sensitive to the type and set-up of the parameterization

scheme. For orographic GWs, their parameterized forcing in the extratropical lower stratosphere, the so-called valve layer, affects the resolved wave propagation from the troposphere to the stratosphere in both hemispheres, influencing the middle atmospheric winter circulation in the models (Šácha et al., 2021; Hájková and Šácha, 2024) and can also control the frequency of Sudden Stratospheric Warmings (Sigmond et al., 2023). In addition, both parameterization types influence the transport characteristics in the middle atmosphere of the models (Eichinger et al., 2020).

The statistical features, spatial distribution, short and long-term variability of the parameterized gravity wave drag were studied extensively (Šácha et al., 2018; Kuchař et al., 2020). Yet, little is known about the gravity wave drag in the real atmosphere, because conceptual difficulties emerge, when trying to constrain the gravity wave drag from observations. Generally, constraining the GW parameterizations is complicated, because this requires knowledge of the global distribution of GW activity and characteristics, i.e. wavelengths, frequencies, momentum fluxes and their divergences, which cannot currently be derived due to the lack of suitable global-scale observational data. Instead of the drag itself, various wave activity proxies are generally derived (see e.g. a short survey in Achatz et al., 2024). Moreover, due to limited observational capabilities to monitor the middle and upper atmosphere, any notion of observationally derived climatologies or trends of dynamical variables in these regions is elusive and likely remains so in the foreseeable future (Añel et al., 2025). Given the importance of GW parameterizations in modeling circulation and dynamics and their tuning for present-day conditions, the lack of understanding of how GWs respond to changes in the climate state introduces a major source of uncertainty in Earth system model projections of the future (Achatz et al., 2024).

With the development of numerical weather prediction and advanced data assimilation and computing techniques, modern global reanalysis datasets start resolving parts of the atmospheric mesoscale spectrum (Li et al., 2023). Although numerous factors like resolution, numerics of the forecast model, parameterization schemes and assimilation methods influence the resulting GW fields, recent studies have already documented the suitability of reanalyses for studying GWs and their effects (including GWD) in the real atmosphere (Podglajen et al., 2020; Gupta et al., 2021, 2024; Pahlavan et al., 2023, 2024; Lear et al., 2024). In particular, the latest version of ERA 5 (Hersbach et al., 2020) has been used extensively for GW research and its validation with observations has shown that the GW representation is realistic, though the amplitudes are considerably underestimated (Gupta et al., 2024), which also translates to the underestimation of GWD in ERA 5.

In this paper, we use 44 years of the ERA 5 dataset to study the vertical distribution and short- and long-term variability of resolved GWD, both in the zonal mean perspective and also regionally averaged above selected GWD hotspot regions. For the first time in the literature, thanks to the advanced methodology, we do so by accounting for the divergence of all of the Reynolds stress tensor components, hence fully respecting the three-dimensional nature of the GW propagation and breaking. The novelty of this computationally demanding research effort is underlined by the fact that, compared to the studies Gupta et al. (2021, 2024), we compute the momentum flux divergences from the full model-level data, which ensures unprecedented accuracy of the presented GWD estimates.

The manuscript is structured as follows - in Section 2, the methodology of gravity wave filtering, computation of drag and the statistical analysis of the obtained time series is described. In Section 3 we present an analysis of the vertical distribution

and of the seasonal cycle of GWD in a zonal mean and on subdomains, followed by an analysis of statistical properties and interannual variability of the GWD. Finally, a discussion and conclusions are presented in Section 4.

## 2   Methodology

We base our study on hourly ERA 5 data on model levels with the horizontal resolution corresponding to about 31 km (0.28125 degrees) for the period 1979 – 2023 (Hersbach et al., 2017), combined with ERA 5.1 data, which correct stratospheric temperature bias present in ERA 5 for years 2000 – 2006 (Simmons et al., 2020). The computation was performed on 31 model levels between 70 and 5 hPa, where the vertical hybrid levels coincide with the pressure levels. This range was extended further down by interpolating ERA 5 data to 5 pressure levels, logarithmically spaced between 120 hPa and 70 hPa using linear interpolation. A weak sponge layer in ERA 5 attenuates the gravity wave perturbations above 10 hPa, with a stronger sponge layer above 1 hPa (Pahlavan et al., 2023). Since the gravity waves are influenced mostly by the top sponge layer (Gisinger et al., 2022), we analyze also a region slightly above 10 hPa.

For calculating the momentum fluxes and drag (momentum flux divergences) due to gravity waves, first, horizontal velocity perturbations due to GWs are separated, as described in Sec. 2.1. We do not apply any filtering of the vertical velocity field as the theory and existing literature (Sun et al., 2023) suggests the dominance of the GW perturbations to the leading order and a filtering procedure might possibly introduce some artifacts to the resulting fields.

The drag computation is similar to the method applied in Kruse et al. (2022) or Procházková et al. (2023) with small modifications (see Appendix A) for computation in pressure coordinates and also for the zonal mean drag estimates, where averaging over latitudes instead of the averaging over subdomains is used. The major difference between the computations of the fluxes and drag in pressure and $z$-coordinates is the assumption on the density. In the $z$-coordinates, it is assumed in the methodology of Kruse et al. (2022) and Procházková et al. (2023) that the density is constant at the vertical levels. In the pressure coordinate, a weaker assumption that the density $n = -\rho g \partial z / \partial p$ is constant at pressure levels can be made (here, $g$ is the gravity, $z$ is the altitude and $p$ is the pressure). To save computation time, the computations on model levels are performed with a realistic assumption $n \equiv 1$, corresponding to the hydrostatic equilibrium. For the sake of completeness, the derivation is shown in the Appendix A.

Two averaging methods are used to evaluate drag: zonal averaging and region-based averaging. Regionally, we evaluated the momentum flux divergences at five different subdomains, Himalayas (20 – 40° N, 70 – 102.5° E), East Asia (30 – 48° N, 110 – 145° E), West America (27.5 – 52° N, 235 – 257.5° E), northern part of Southern Andes (30 – 45° S, 283 – 305° E) and southern part of Southern Andes (45 – 60° S, 283 – 305° E), as displayed in Fig. 1. The subdomains must be sufficiently large to accommodate the GW wavelengths while also being small enough to prevent alterations in the synoptic wind.

### 2.1   GW filtering methods

In accordance with the approaches applied in recent literature (e.g., Gupta et al., 2024; Reichert et al., 2024), we separate the GW perturbations using the horizontal spherical harmonic filtering. In this type of methodology, the data are transformed into

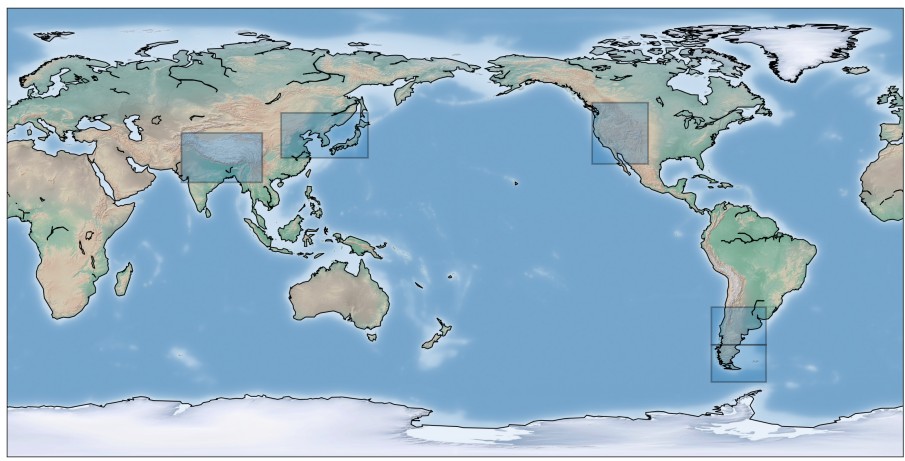

**Figure 1.** Map with the selected subdomains: Himalayas, East Asia, West America and two subdomains at Southern Andes are highlighted.

spherical harmonics and amplitude of the modes with wavelengths exceeding a specified limit are set to zero. Since the cut-off that distinguishes mesoscale gravity waves from synoptic-scale waves varies with time and region (Procházková et al., 2023), it cannot be optimally diagnosed in global analyses and must be determined a priori.

Very often, the exact implementation of the scale separation is based on a so-called triangular truncation of the spherical harmonics, which sets a maximal admitted total wavenumber as the cut-off. For the studies of GWs, total wavenumber of 20 (Dörnbrack et al., 2022; Polichtchouk et al., 2022) or 40 (Wedi et al., 2020; Wei et al., 2022) is usually taken as the limit, selecting the maximal wavelength of GWs as approximately 2000 or 1000 km for both zonal and meridional direction, respectively. However, if the wave is not purely zonal or purely meridional, limiting the total wavenumber in the triangular

truncation also admits modes with much larger wavelengths (see left part of Fig. 2). To avoid this issue, we implement a method based on the rhomboidal truncation (Daley and Bourassa, 1978).

     As illustrated in Fig. 2, instead of taking a maximal total wavenumber, the rhomboidal method is based on two limits: a limit $M$ on the absolute value of the zonal wavenumber $|m|$ and a limit $N$ on the difference between the total wavenumber $n$ and the absolute value of the zonal wavenumber $n - |m|$ (altogether, we have $m \leq M$ and $n - |m| \leq N$). The rhomboidal truncation

is used for instance in spectral numerical weather prediction models (Vasylkevych and Žagar, 2021) and it was also used in a few recent studies as a filtering method to separate synoptic waves (Xie et al., 2022; Frederiksen and Frederiksen, 2011).

     For proximity with the typical triangular truncation limits used in the recent literature mentioned above, we apply the rhomboidal method with $M = 20$ and $40$ and $N = M - 1$. We will refer to these methods as Rho20 and Rho40 in the paper. The use of two different cut-offs enables us to distinguish between the effects caused by resolved waves with shorter and longer

horizontal wavelengths.

     The zonal wind filtered using the rhomboidal truncation is displayed in Fig. 3 for one time instant at 50 hPa together with the (unfiltered) vertical wind for comparison. As a proof of concept, we see similar patterns and scales for the filtered zonal velocity as for unfiltered vertical velocity (e.g., Southern Andes or West America orographical hotspots and numerous non-

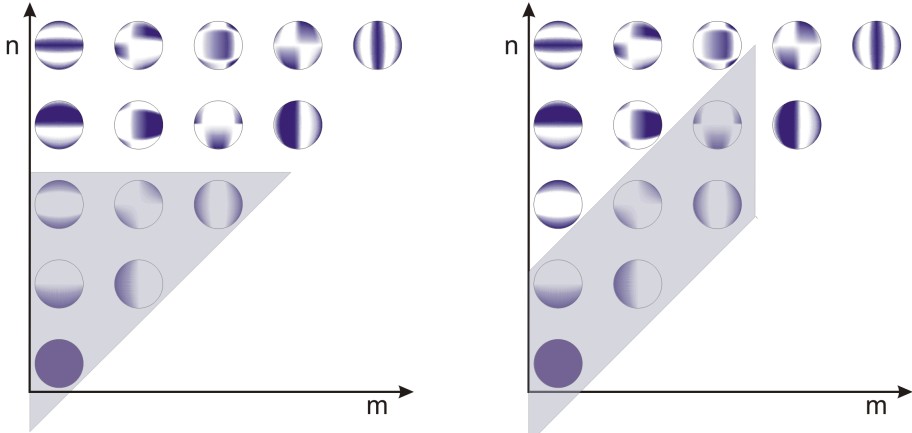

**Figure 2.** Scheme of the triangular (left) and rhomboidal (right) truncation of spherical harmonics with the total wavenumber $n$ and the zonal wavenumber $m$. In the gravity wave filtering, amplitudes corresponding to the shaded modes (here, triangular truncation with the total wavenumber 2 and the rhomboidal truncation with the limits $N = 1$ and $M = 2$) are removed from the data. Only parts with positive $m$ are displayed.

orographic GW signatures above the ocean and tropics). A perfect match between vertical velocity and horizontal velocity perturbations cannot be expected, as polarization relations indicate that gravity waves with shorter horizontal wavelengths tend to have higher ratio of vertical to horizontal wind amplitudes and vice-versa (see Eq. 36 in Fritts and Alexander, 2003). In line with this, the similarity between the vertical and horizontal wind perturbations is enhanced for finer filtering (Rho40).

At first glance, comparing the rhomboidal and triangular truncations (Figs. 3a and 3c) reveals similar perturbation patterns, although the triangular truncation exhibits generally higher amplitudes. This follows from the definition of the two filtering methods, where rhomboidal truncation removes certain longer-scale modes that would be retained under triangular truncation. A more detailed view reveals also differences in details of the GW patterns, especially in the high latitudes due to the anisotropy of the rhomboidal method. Note for example the region to the west of New Zealand, where the triangular truncation keeps perturbations in the meridional direction that are mostly removed by the rhomboidal truncation, leaving only a more clear small-scale structure consistent with the vertical wind in Fig. 3d. In general, while the rhomboidal truncation in ERA 5 slightly increases the underestimation of GW amplitudes, we argue that it improves the accuracy of GW field detection from a methodological standpoint.

## 2.2 Multiple linear regression

Multiple linear regression (MLR) is a standard approach to assess long-term trends of atmospheric time series (e.g., Zhao et al., 2021) incorporating the internal variability of the climate system, which can be, to some extent, understood as a superposition of quasi-periodic oscillations such as El Niño–Southern Oscillation (ENSO), North Atlantic Oscillation (NAO), solar cycle, quasi-biennial oscillation (QBO) etc. For our analysis, we used MLR with seven predictors: Multivariate ENSO Index Version

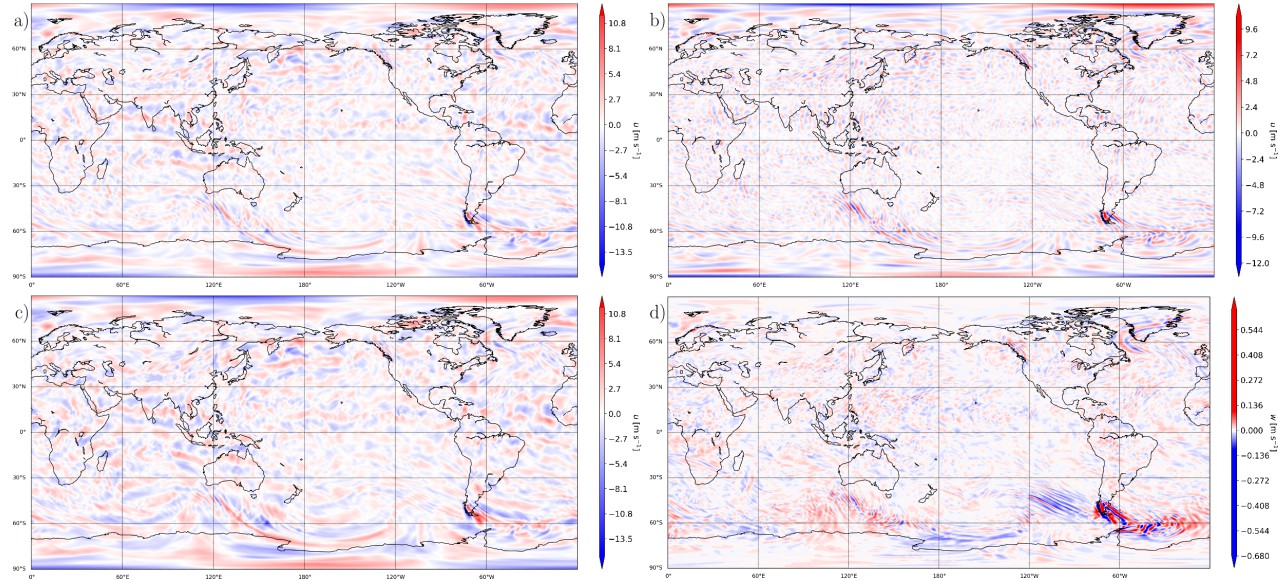

**Figure 3.** Filtered fields of the zonal wind velocity at 50 hPa for the Rhomboidal method with (a) $M = 20$, (b) $M = 40$, (c) Triangular method with the total wavenumber 20 and (d) unfiltered vertical wind velocity.

2 (MEI.v2), NAO index, monthly mean total sunspot number, and four QBO indices based on zonal mean wind at 10, 30, 50 and 100 hPa. Summary of the indices used in our study can be found in Table B1. Due to the indices availability, we consider only the data between 1979 and 2021 for the MLR analysis.

For a straightforward interpretation of the obtained regression coefficients, values of all the indices were normalized to the amplitude of unity over the time period. Furthermore, the linear trend was removed from all indices using the ordinary least squares method. Statistical significance was calculated for each regression term using a two-sided Student's t-test. The MLR analysis was applied on seasonal averages, which follow the normal distribution in the discussed regions.

## 3 Results

### 140 3.1 Vertical distribution of the climatological GWD

In the first step, we present the seasonal climatologies of the zonal mean net zonal and meridional drag (GWDx and GWDy), as displayed in Fig. 4. For all seasons, we see localised maxima of the negative zonal drag in the extratropics at the upper flank of the subtropical jet stream (seasonal climatology of the zonal mean zonal wind is shown in contours). This is a very important result for many reasons. The localized zonal mean zonal drag maxima above the upper troposphere and lower stratosphere

(UTLS) jet are a robust feature across orographic gravity wave parameterizations in climate models (e.g., Hájková and Šácha,

2024). It is a direct consequence of the saturation criterion employed in the parameterizations (Lindzen, 1981; Nappo, 2012) that requires momentum flux convergence in the region of negative wind shear above the jet center, where the increase of the mean wind does not balance the decrease of the density anymore. This follows from an estimate of the maximum momentum flux that can be propagated vertically without breaking for a given zonal wind profile. Assuming motions confined to the $x$–$z$ plane, let $\overline{u}$ represent the vertical profile of the mean wind in the $x$-direction, $k$ the horizontal wavenumber, $c$ the phase velocity, $\overline{\rho}$ the mean density profile and $N$ the Brunt-Väisälä frequency. Under these assumptions, the upper limit on the momentum flux can be expressed as follows (Fritts, 1984):

$$MF^s_{zx} = -\overline{\rho}\frac{1}{2}\frac{k}{N}\left(\overline{u}-c\right)^3, \tag{1}$$

which highlights the dependence of the maximum momentum flux on the background wind profile.

Reproducing the negative wind shear and separating the UTLS and polar night jet was one of the goals and historical motivations for implementing orographic GW parameterization in the models. The robust representation of this feature in the resolved GWD across both hemispheres offers a dual benefit: it reinforces confidence in the realism of our GWD estimates and provides an atmosphere-based validation of a key assumption used in climate model GW parameterizations. Furthermore, this region has been identified in previous coarser versions of reanalyses as a local maximum of the residual term of the zonal mean zonal momentum budget (Fujiwara et al., 2024). This residual term is supposed to be caused by a sum of the parameterized GW forcing and missing GW forcing likely included in the assimilation increment (Sato and Hirano, 2019). The collocation with a region of maximal resolved drag presented here supports this hypothesis that the residual term in the momentum budget is linked to GWs.

Regarding the magnitudes of the zonal mean zonal drag in the lower stratosphere, they are higher in the winter hemisphere, which is connected with the stronger circulation and momentum fluxes from below. Further upward in the winter hemisphere, the negative zonal mean zonal drag region is focused more poleward towards the polar night jet and the magnitude again increases and would peak in another maximum above the upper boundary of our analysis, at the upper flank of the polar night jet (Eichinger et al., 2020). In the tropics and summer hemisphere extratropics, we can see regions of weaker positive zonal mean zonal drag decelerating the climatological easterlies.

The zonal mean meridional drag component has not received much attention in the literature so far, although it is also produced by the GW parameterization schemes and can have profound dynamical effects in the models (Šácha et al., 2016). Using Ray tracing simulations, Kalisch et al. (2014) demonstrated that the occurence of meridional GWD is predominantly tied with the GW oblique propagation. In their work, the meridional drag was pronounced mainly in the upper stratosphere and mesosphere and it was mostly directed poleward. While a poleward meridional drag dominates in the summer hemisphere, its direction varies considerably elsewhere, exhibiting particularly sharp changes within the polar night jet. Stronger meridional drag magnitudes can be seen also at the upper flank of the UTLS jet. Here, the drag tends to decelerate the mean meridional circulation equatorward from the jet center and accelerate it poleward. How realistic this feature is (and the sharp variations inside the polar night jet) and whether it is connected also with the oblique propagation remains yet to be understood.

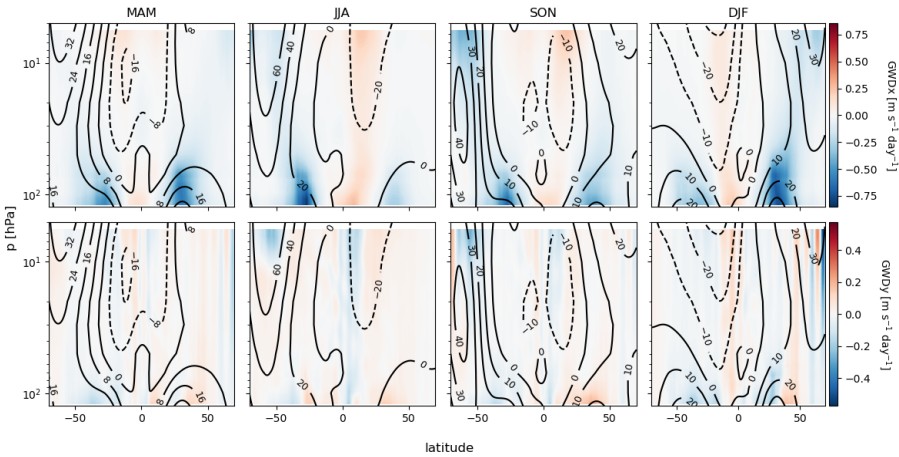

**Figure 4.** Climatology of the zonal mean directional drag for individual seasons obtained using the Rho20 method. Contours show the zonal-mean zonal wind in ms$^{-1}$.

## 3.2 GWD hotspots in the lower stratosphere

We have seen in the previous section that on the vertical domain of our analysis the GWD is strongest in the lower stratosphere. From global observations it is known that the GW activity is asymmetrically distributed along the globe in so-called hotspots (Hoffmann et al., 2013; Šácha et al., 2015). Our methodology allows us to compute regional drag estimates, which we do in the following for selected major extratropical hotspots, evaluating also the effects of horizontal fluxes. Summing all the contributions together (vertical and horizontal flux divergence components), we have a time series of a net GW drag above the

hotspots in the zonal or meridional direction for each vertical level.

We start by analyzing the temporal variability of the drag in the zonal direction. The frequency spectra of GWDx for the 70 hPa level, coinciding with the drag maximum in the zonal mean, are shown in Fig. 5. The spectra have significant noise, a few peaks are, however, still distinct for all subdomains. The first, highest peak marks the yearly cycle of the drag. Connected to that are also modulating periods of a half and a third of a year, also visible in the spectra. Additionally, we see the spectral

power enhanced for the period of one day and the shorter periods of 12, 8, 6, 4 and 3 hours, pointing towards the existence of a daily cycle in GWD. This can be caused either directly by solar heating of the air and the Earth's surface or secondarily by coupling with the solar tides. In Figure 5, we also illustrate the corresponding value of inertial frequency. Nevertheless, since the inertial frequency is close to the daily periods on the subdomains and since we study the spectra for the extrinsic frequency, we do not see any clear relation to enhancements in the spectral power.

The yearly cycle is a dominant mode of variability for GWDx and its seasonality above hotspots is further documented in Fig. 6 for the 70 hPa level. The figure demonstrates the sensitivity of the regional GWD estimates on the choice of the exact hotspot location and illustrates our motivation for splitting the Andes hotspot into two. Theory, previous zonal mean results and regional drag estimates suggest a distinct annual cycle of zonal GWD in the extratropics, particularly in hotspots associated

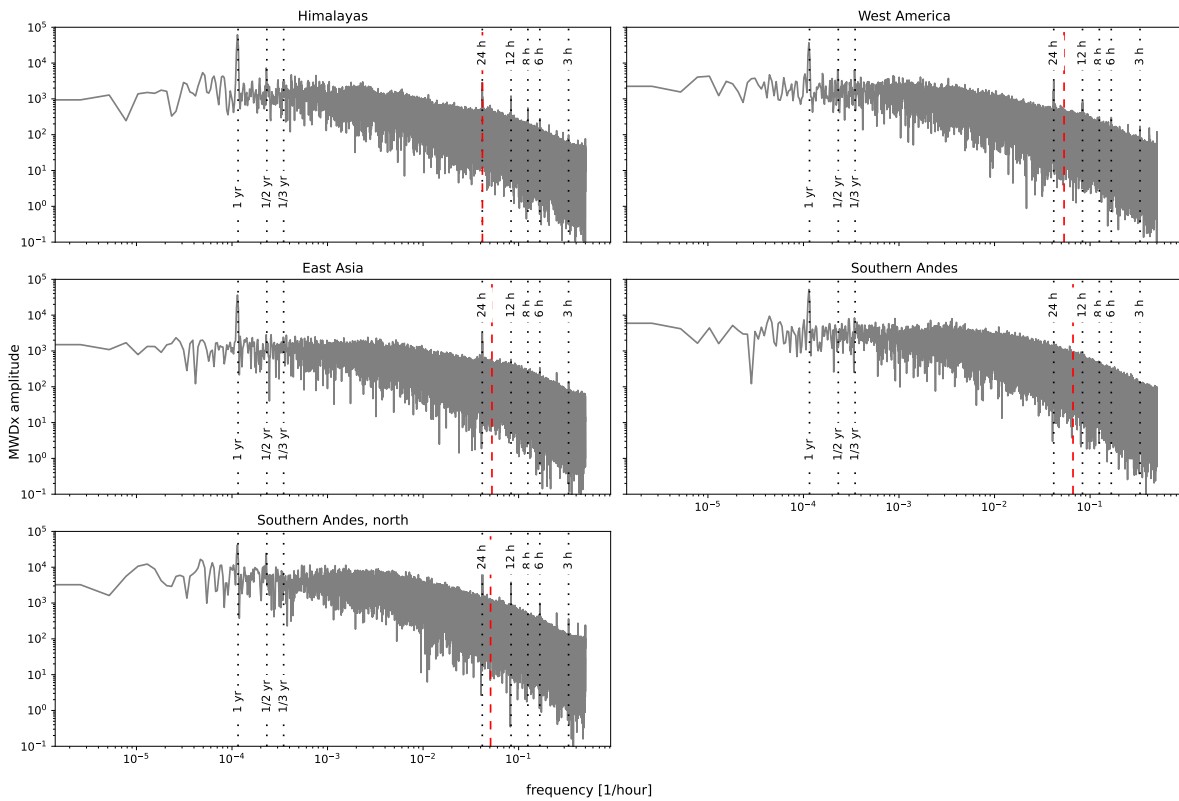

**Figure 5.** Frequency spectrum of GWDx for the level 70 hPa computed using the Rho20 method. The black dotted lines from left to right denote the periods one year, half year, third of a year, one day, 12 hours, 8 hours, 6 hours, 4 hours and 3 hours. The red dashed line is the inertial frequency.

with significant orography. Indeed, in the studied Northern Hemispheric hotspots, we see a pronounced yearly cycle of the drag
with minima (strongest drag) around February and maxima around August. For the Southern Andes regions (as shown in Fig. 1), two opposing yearly cycles can be derived. The seasonal cycle for the northern Southern Andes subdomain follows the same rule as for the hotspots in the Northern Hemisphere - with maxima in March (late Southern Hemisphere summer) and minima in July (Southern Hemisphere winter), however, for the southern part of the hotspot, the cycle has an opposite phase (overlapping with the Northern Hemisphere hotspots in the figure). The reason for that is a different vertical profile of zonal wind above the
subdomain in Southern Hemisphere winter, with the subdomain being situated already poleward from the UTLS jet with winds continuously increasing with height toward the polar night jet. Therefore, during the Southern Hemisphere winter, the vertical profile of the zonal wind above the subdomain does not suggest any regions of potential instability or critical level filtering according to the saturation hypothesis and the resolved wave field behavior confirms this. For other hotspots, we observe the signature of the UTLS jet in all seasons (see Fig. B1).

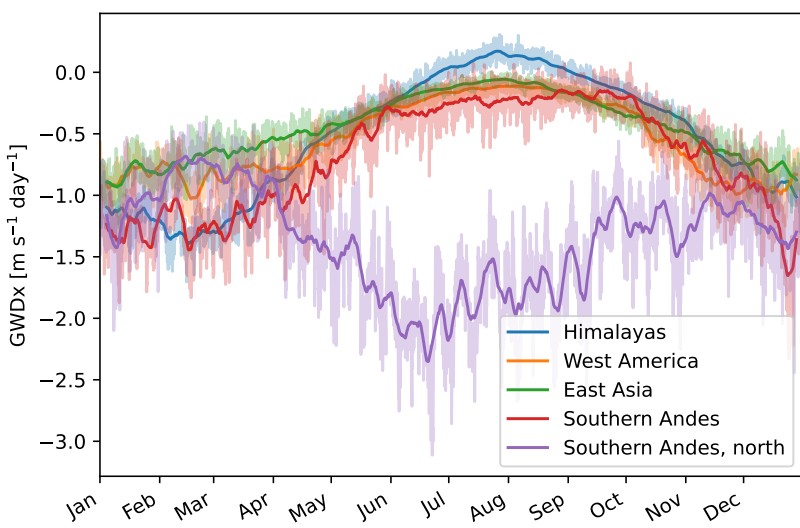

**Figure 6.** Seasonal cycle of GWDx above hotspots at 70 hPa using the Rho20 method. The solid line displays 7-day running mean.

The vertical zonal drag profiles over the hotspots during their peak drag season (Fig. 7) clearly illustrate the significance of the vertical zonal wind profile, particularly highlighting the presence of a negative wind shear region above the center of the UTLS jet. As a result, we can see localised drag maxima around 70 hPa all hotspots. This again documents the robustness of the saturation criterion for parameterizations of GW effects. Interestingly, for the West America hotspot the maximum is not sharply localized, but the upper flank of the UTLS jet rather marks an end for the strong drag across the UTLS. We can only speculate at this point, whether this is connected with comparably stronger momentum fluxes reaching the instability earlier during their propagation from below or with a more significant presence of other sources than orographic GWs, including possibly downward propagating modes. The vertical profiles do not deviate significantly from profiles that would be obtained by the filtering with triangular truncation (see Fig. B2).

Figure 8 shows the vertical profile of the horizontal drag component caused by the zonal derivative of the zonal flux of zonal momentum (GWDxx). It contributes approximately 10 percent to the climatological GWDx magnitude and exhibits a similar vertical profile to the dominant GWDx component, the vertical derivative of the vertical flux of zonal momentum (GWDzx), which can be inferred as the main part of GWDx in Fig. 7. For completeness, the GWDyx drag component is shown in the attachment in Fig. B3.

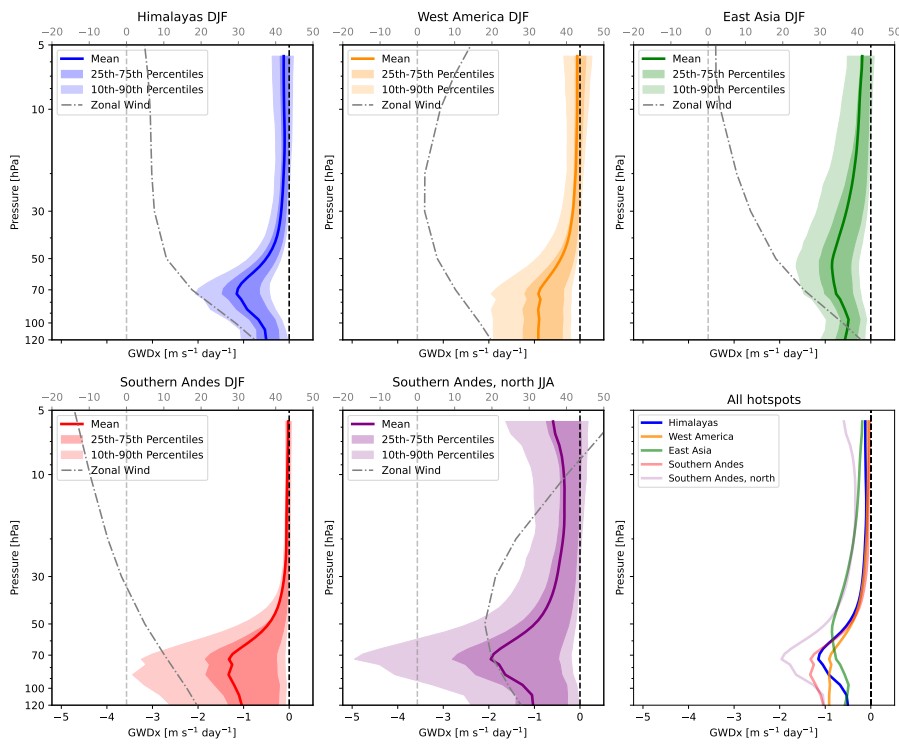

**Figure 7.** Climatological vertical distribution of GWDx above the hotspots for their respective peak seasons, obtained by Rho20 method.

## 3.3 Statistical properties of GWD

From observations (Wright et al., 2013), we know that the GW activity is highly intermittent and this is also the case for the parameterized GWD (Kuchař et al., 2020). Figure 9 shows the probabilistic distributions of hourly values of the resolved GWDx at 70 hPa for each hotspot during its peak season (JJA for northern part of Southern Andes, DJF for other subdomains). Although we have seen that the mean climatological GWDx is negative, the histograms reveal also a portion of positive drag events, a feature that is almost completely missing in parameterizations (compare with Fig. 6 of Kuchař et al., 2020), where

the probabilistic distribution resembles the log-normal distribution. For the resolved drag, the distribution is closer to a skew normal distribution, which is asymmetrical around zero, with much more power and long tails for negative GWD values.

The probability distributions for different scale-separation cut-offs in Fig. 9 also reveal different contributions to the regional GWDx values by shorter and longer waves. We can clearly see that the inclusion of longer waves leads to increased probability of strong negative GWDx and its extremes robustly across the hotspots. Limiting the analysis to the Rho40 method only would

result in substantial reduction of the drag intermittency and extremity especially for Himalayas and the northern subdomain of Southern Andes.

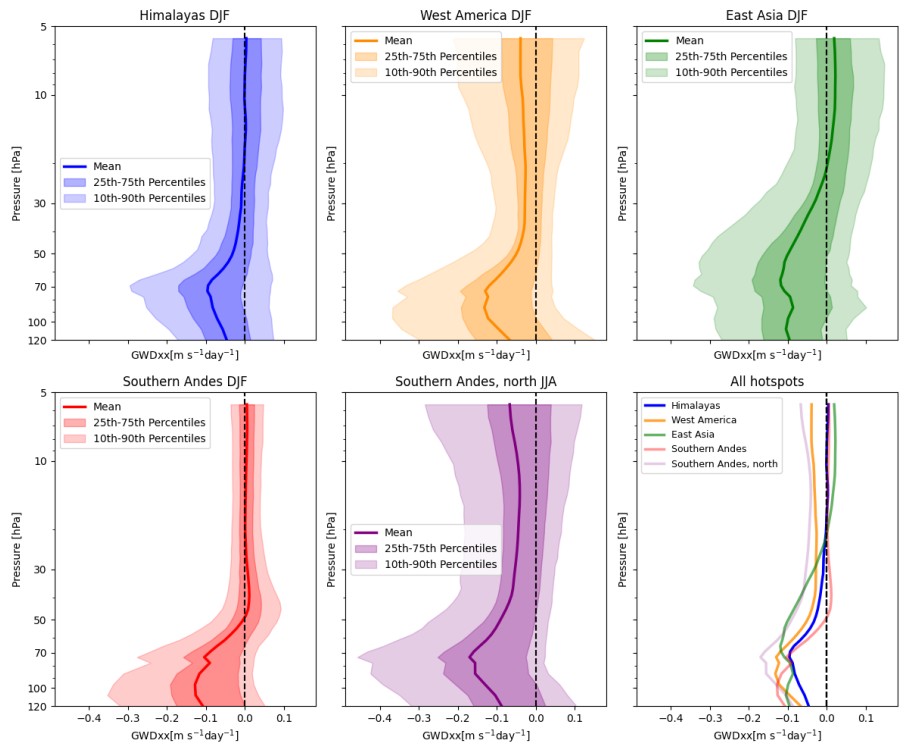

**Figure 8.** Climatological vertical distribution of GWDxx above the hotspots for their respective peak seasons, obtained by Rho20 method.

Figure 10 allows us to analyze how the horizontal momentum flux divergences (connected with oblique propagation) contribute to shaping the probabilistic distribution of the net zonal GWD magnitudes and whether the mechanism of contribution differs for the longer and shorter waves. Based on linear theory, we anticipate that the horizontal components (GWDxx,
GWDyx) of the zonal drag will be less important for shorter waves. This is indeed visible for the West America and East Asia hotspots. Although the GWDyx component is relatively small, the GWDxx component obtained by the Rho20 method is comparable in magnitude with the GWDzx component and significantly contributes to the occurence of strong zonal drags at the hotspots. Interestingly, for the Himalaya hotspot, longer wave modes do not contribute as strongly to the horizontal flux divergences, but contribute strongly by the means of vertical flux divergence. A possible reason for this can be that the horizontal
scales and geometry of the Himalaya orography together with its orientation with respect to the predominantly zonal background flow of the Himalaya hotspot favor sourcing of longer orographic GW modes that propagate vertically more efficiently compared to longer GWs in other hotspots. For both subdomains at Southern Andes (Fig. B4 in the appendix), the contribution from horizontal components to GWDx is smaller than from the vertical component and they mostly do not contribute to longer tails of the spectrum (although their contribution to the mean drag is not negligible - about 1/3 of the mean zonal drag from the
vertical component for JJA and about 1/6 for DJF, not shown).

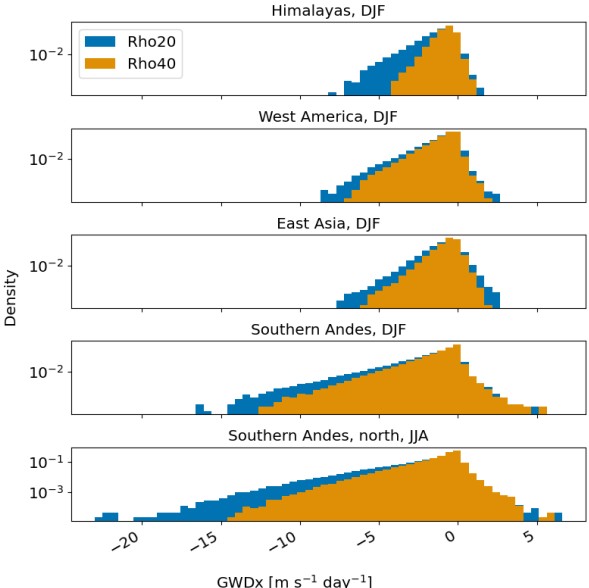

**Figure 9.** Histograms of GWDx for peak seasons of GWD at 70 hPa for the hotspots. Note that the probabilistic density is plotted with a logarithmic axis.

## 3.4 Interannual variability and trend of GWD in the lower stratosphere

In this subsection, we explore how GWs, specifically the drag effects in the lower stratosphere, respond to changes in climate conditions. We begin by analyzing the long-term variability of the zonal mean zonal drag components. Figure 11 shows the long-term trend obtained from the MLR for DJF and JJA.

In the extratropical regions, the traditional MLR can explain just a minor part of the interannual GWD variance ($R^2$ around 20 %). Nevertheless, we see statistically significant weak positive drag trend at various pressure levels in the mid-latitudes summer hemisphere. On the winter hemisphere, the significant trend is mostly negative with stronger positive trend in higher altitudes around the polar vortex. Due to the mostly negative zonal drag in extratropics, positive (negative) trend in GWDx implies decrease (increase) of the drag magnitude. Therefore, GWs tend to exert higher GWDx amplitude in the winter hemi-
sphere and a weaker drag in the summer hemisphere and near the polar vortex.

    For the lower stratosphere, we see a significant positive trends around the upper flank of the UTLS jet, where we have diagnosed a maximum in the zonal mean GWD in the first section. This is a clear fingerprint of the structural changes in the atmosphere (both in horizontal and vertical structure) influencing and explaining the significant GWD trends in this region. Namely, with the thermal expansion of the troposphere due to increasing greenhouse gas (GHG) levels, the tropopause pressure
decreases, i.e., the tropopause rises relative to the ambient pressure levels, and the circulation in the vicinity (UTLS jet) tends to follow it. This acts to shift also the critical levels for GW propagation and breaking and hence the local GWD maximum upward. This upward shift projects into the significant weakening (positive trend) of GWD from 120 to 90 hPa at the equatorward side

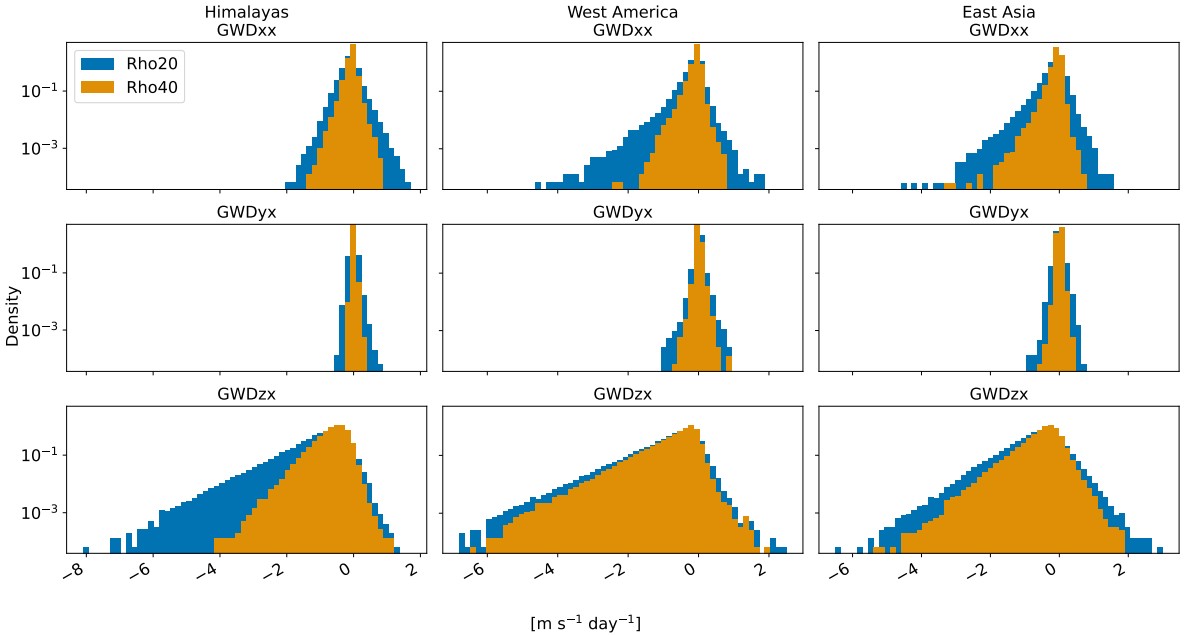

**Figure 10.** Histogram of GWDx for DJF on the Northern Hemisphere subdomains. The probabilistic density is plotted with a logarithmic axis.

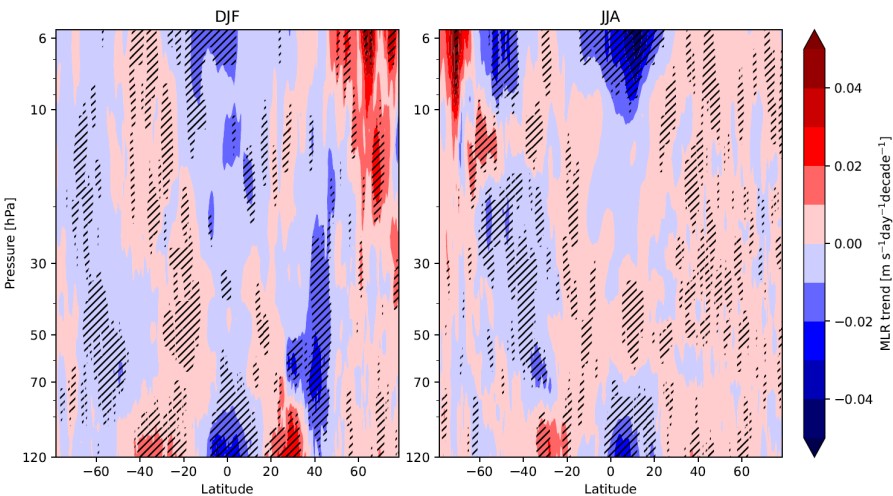

**Figure 11.** MLR trend for zonal mean GWDx for DJF and JJA, obtained by Rho20 method. Statistically significant trends at a confidence level of 95 % are hatched.

of the jet and strengthening (negative trend) from 90 hPa upwards, which is slightly more pronounced at the poleward side of the jet (especially from about 90 to 70 hPa). This pattern can be seen for the winter hemisphere for both solstice seasons.

In the equatorial regions between 20° S and 20° N, the GWD variance can be well explained by the selected regressors ($R^2$ around 80 %). The MLR analysis reveals strong significant negative trends in GWDx for both lower and upper stratosphere, corresponding to the increase of GWDx. However, due to the complicated situation on the equator with combinations of eastward and westward modes, we refrain from drawing conclusions regarding the meaning of the possible drag changes in this region. A dedicated study would be needed for this.

Figure 12 depicts selected regressors of the MLR with significant signal NAO, ENSO and QBO at 100 hPa (QBO100). As for the NAO, the oscillation significantly affects the GWs mostly on the Northern Hemisphere in winter, increasing the GWDx at all pressure levels at about 40° N and decreasing it at 60° N and poleward, which suggests poleward shift of the GW dissipation connected with the changes of the circulation. Similarly, we see a significant influence of the ENSO on the GWDx, causing negative anomalies for both summer hemispheres, although being much stronger on the Southern Hemisphere. This represents intensification of the zonal GWD component during the warm, El Niño period. The intensification is consistent with the ENSO response between 20° and 40° S found for example from SABER measurements in Ayorinde et al. (2024) or from model simulations in Simpson et al. (2011).

The rightmost column in the plot shows the QBO100 regressor for DJF and JJA. For both seasons, it causes in the equatorial region a decrease of GWDx magnitude between 40 and 20 hPa and an increase of the magnitude further up. In the extratropics, the QBO winds at 100 hPa are connected with significant differences in the GWDx in the upper layers of polar vortex, causing a poleward shift of drag during westward QBO winds. In addition, it causes stronger GWDx maxima above the UTLS jet. Nevertheless, a two-way interaction between GWs and QBO has to be anticipated, which adds uncertainty to the analysis due to the nonlinear effects that are not contained in the MLR analysis.

Apart from the shown regressors, influence of QBO winds at 10, 30 and 50 hPa and the total sunspot number were examined. The QBO regressors mostly affect the drag in the tropics by causing drag decrease in the lower stratosphere and increase in the upper stratosphere. However, due to the complexity of GW fields in the tropics and limitations due to nonlinear interactions with QBO mentioned above, we do not discuss these results further. As for the solar activity, we do not see any clear and significant signal emerging in GWDx.

The results in Figs. 11 and 12 are obtained using the Rho20 method. However, if we limit the analysis to the shorter waves only (Rho40 method), we obtain very similar results (not shown).

Regionally, for the hotspots, contrary to the results for parameterized drag by Šácha et al. (2018), the imprint of climate oscillations from MLR and the trends are seldom significant (not shown). For the trends during respective winter seasons, we see mostly decrease of the drag value (increase of the magnitude) around 70 – 40 hPa and increase below, which is consistent with the changes around the UTLS jet described above. The trend is however significant only for specific hotspots. For example, it is significant for Himalayas DJF GWDx increase. For some subdomains, it is significant for a part of the spectrum only (e.g., DJF GWDx decrease for Himalayas only for Rho20 or JJA GWDx decrease for Southern Andes for Rho40 only), without any clear systematic signal. Similarly, the significance for some subdomains differs for GWDx and its vertical component GWDzx, although the vertical profile is very similar, again without any clear pattern of a trend scenario. We cannot say if, for example, longer waves and inclusion of horizontal propagation effects contribute positively or destructively to the regional trends.

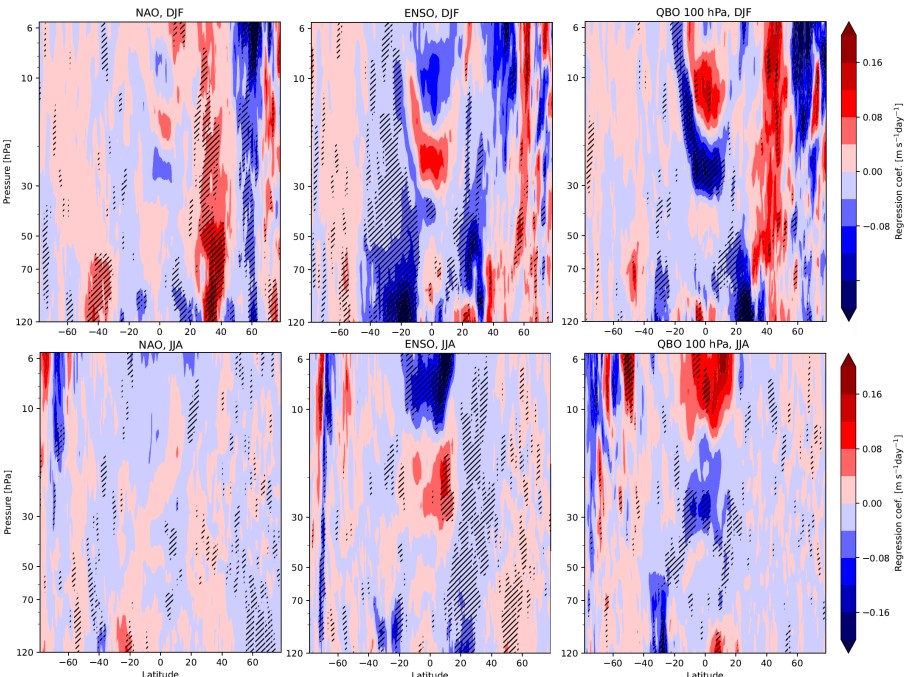

**Figure 12.** Selected regressors for zonal mean GWDx for DJF and JJA, obtained by Rho20 method. Statistically significant regressors at a confidence level of 95 % are hatched.

Although we did not observe a consistent, statistically significant trend in the net zonal drag above the hotspots, we note changes in the GWD distribution when comparing 22-year periods before and after the year 2000 (see Fig. 13), i.e. in the ozone depletion and recovery periods. In all subdomains analyzed, we find a higher frequency of occurrence of small to no drag events (close to zero or zero GWDx) during the earlier period, whereas in the period after 2000 we can see more frequent events with substantial GWDx. This change in GWDx intermittency may be connected with regional circulation changes associated with ozone depletion and recovery, but may also reflect a specific fingerprint of the vertical shift due to increasing GHGs in the circulation and drag changes.

## 4 Discussion and conclusions

This study presents a first analysis of resolved gravity wave drag climatology, variability, and trends in the stratosphere underlaid by a realistic atmospheric state, circulation and their evolution in the ERA 5 reanalysis. The results are presented in the zonal mean framework and also regionally for selected GW hotspots. The zonal mean drag climatology in the stratosphere exhibits a vertical distribution consistent with the saturation hypothesis, as predicted in simplified models, featuring pronounced GWD maxima above the center of the UTLS jet across all seasons in both hemispheres. Also, the meridional drag component, GWDy, is maximal in this region, providing evidence for the importance of oblique GW propagation already in the lower

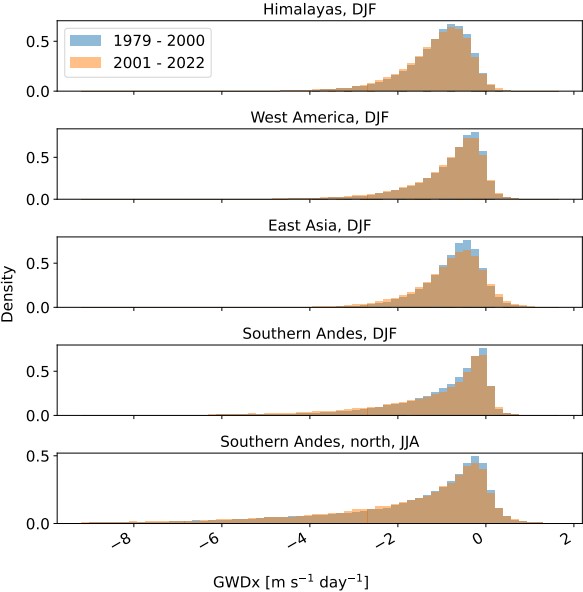

**Figure 13.** Histograms of GWDx for peak seasons of GWD at 70 hPa for the hotspots comparing the periods 1979 – 2000 and 2001 – 2022, obtained by the Rho20 method. The range of the horizontal axis does not correspond to the range of the data.

stratosphere. The drag maximum above the center of the UTLS jet is reflected also in the regional drag climatologies above the
hotspots. Mean zonal drag magnitudes in the peak season reach few m s$^{-1}$ day$^{-1}$, but analyzing the probability distributions at 70 hPa reveals extreme events with drag magnitudes around 5-10 m s$^{-1}$ day$^{-1}$ (for Andes also exceeding 20 m s$^{-1}$ day$^{-1}$) above the hotspots in the lower stratosphere. Lateral propagation is shown to contribute significantly (around 10 %) to the net zonal drag regional climatology and the relative contribution can be much greater for extreme drag events.

We provide also novel insights into the variability of the lower stratospheric GWD. Above the hotspots, annual and daily
cycle are dominating the short-term variability of the zonal drag and for the zonal mean zonal drag, we have found significant imprints of NAO, ENSO and also QBO signals in the extratropical stratosphere. We also report significant long-term trends of GWDx, both zonal and regional mean that are most likely driven by the tropospheric expansion due to increasing GHG emissions and upward shift of the UTLS jet. More to this, for the hotspots, we have found indications that the extremity of GWDx have changed before and after the year 2000, which could be in addition to GHGs also connected with background
circulation changes between the ozone depletion and recovery periods.

Generally, we can say that the novel GWD estimates presented here are in a good accordance with the previous understanding of GWD climatology, variability and intermittency based on parameterized drag analysis from nudged climate model simulations (Šácha et al., 2018; Kuchař et al., 2020) or from observational proxies (Ern et al., 2018). Also, the importance of oblique propagation for GWD has been established before (Kalisch et al., 2014; Eichinger et al., 2023; Voelker et al., 2024),
but our results suggest that the horizontal propagation significantly affects also the drag already in the lower stratosphere. The

existence of substantial zonal mean GWDy above the UTLS jet has not been reported before to our knowledge. It has been implicitly assumed that meridional drag can exist locally due to meandering of the background circulation (Šácha et al., 2018), however, the existence of pronounced zonal mean GWDy in the lower stratosphere has not been reported before, even for the parameterized drag. Observational validation of this feature may be one of the tasks for the candidate mission of the ESA Earth

Explorer 11 CAIRT (Rhode et al., 2024). Another question is the possible dynamical impact of GWDy that is very strong relative to the typical scales of the residual mean meridional circulation in the lower stratosphere. Traditionally, in the Transformed Eulerian mean framework the existence of zonal mean meridional drag is neglected. It can form a part of the reasoning why the residual mean circulation in ERA 5 has been found to behave fundamentally differently than in the more coarse reanalyses versions (Šácha et al., 2024).

The analysis of interannual variability of the resolved drag confirmed an expected strengthening of the drag in the lower stratosphere driven by the upward shift and increasing shear of the UTLS jet. This is a reassuring result, because the tropospheric expansion (Match and Gerber, 2022), stratospheric cooling (Pisoft et al., 2021) and resulting changes of the circulation (Lee et al., 2019) are robustly observed and documented features. In addition, we have found pronounced signal of tropospheric climate modes of variability (NAO, ENSO) and of lower stratospheric QBO in the interannual variability of the resolved GWDx

across the extratropical stratosphere. Exact mechanisms and validity of these signal have to be assessed in the future research, but they jointly indicate that resolved GWs act as a mechanism for coupling of the variability between the troposphere and stratosphere and tropics and extratropics and can play a role in dynamical teleconnections between those regions.

The resolved GWD estimates presented here are produced with unprecedented accuracy compared to previous studies (e.g., Gupta et al., 2024), because the flux divergences are computed as accurately as possible - from a model native grid. However, for

completeness, it has to be noted that even so the presented resolved GW drag estimates are inherently connected with substantial uncertainty that, however, cannot be objectively quantified. Throughout the text we have seen that the results are sensitive on the part of the GW spectrum that enters the analysis. In the absence of one universally valid cut-off range, we decided to present our results based on two separation scales frequently used in the literature and by comparing the results between the two cut-off choices, the reader can get an estimate of the uncertainty of the results connected with this choice. The rhomboidal truncation

chosen here for the practical implementation of the scale separation, allows a more elegant implementation of the cut-off (for example the cut-off wavelength due to the limit on the zonal wavenumber slightly increases from the polar regions towards tropics), but has in total only a cosmetic influence on the results. Another type of subjective choice influencing the results is the definition of the domains for the local drag estimates. It has been demonstrated on the example of Andes that two different regional definitions of one hotspot can result in very different drag estimates with different seasonal cycles.

Our study and the study Gupta et al. (2024) present pioneering efforts toward producing quasi-observational estimates with global coverage and long-duration of the drag connected with GWs in the real atmosphere. However, it still needs to be thoroughly validated how realistic the GW fields in reanalyses are. In recognition of this, the A-RIP initiative started a task team dedicated to this effort, as more reanalysis types and versions are starting and will continue resolving major parts of the GW spectrum in near future. International cooperation is clearly needed for this goal, because analyzing resolved GWs is

a demanding task for computational resources and methodological approach. A major boost for putting direct observational

constraints on the GW drag distribution, magnitude and direction would be the launch of the CAIRT mission, which will be specifically designed to allow for the first time to globally observe local GW parameters and induced GW drag (Rhode et al., 2024).

*Code and data availability.* ERA 5 data used for the study are publicly available (Hersbach et al., 2020). For MLR analysis, NAO index was
obtained from https://www.cpc.ncep.noaa.gov/products/precip/CWlink/pna/nao.shtml, QBO data from https://www.geo.fu-berlin.de/met/ag/strat/produkte/qbo/index.html, Multivariate ENSO Index from https://www.psl.noaa.gov/enso/mei/data/meiv2.data. Sunspot number from https://sidc.be/silso/datafiles. Full computed GWD time series can be made available upon request due to their size, a smaller version with daily averages and reduced vertical grid for the Rho20 method can be downloaded at 10.5281/zenodo.15473685. The codes are accessible on Github at github.com/prochazz/era5gws.

**Appendix A:  GW fluxes and drag computation**

**A1    Fluxes and drag on subdomains**

In order to derive formulas for computation of momentum fluxes and drag components caused by gravity waves on subdomains, We start with the horizontal momentum equation in pressure coordinates (Rõõm, 2001):

$$n\frac{\mathrm{d}\boldsymbol{v}_h}{\mathrm{d}t} = -g\nabla z - n\boldsymbol{f} \times \boldsymbol{v}_h, \tag{A1}$$

where $\boldsymbol{v}_h$ is the horizontal velocity, $z$ is the altitude, $g$ is gravitational acceleration, $\boldsymbol{f} = (0,0,f)$ for the Coriolis parameter $f$ and $n$ is the pressure coordinate density defined by

$$n = -\rho g \frac{\partial z}{\partial p} \tag{A2}$$

with $\rho$ being the standard density. The continuity equation has the form

$$\frac{\mathrm{d}n}{\mathrm{d}t} + n\left(\nabla \cdot \boldsymbol{v}_h + \frac{\partial \omega}{\partial p}\right) = 0, \tag{A3}$$

where $\omega \equiv \mathrm{d}p/\mathrm{d}t$.

By combination of equations (A1) and (A3), we get

$$\frac{\mathrm{d}}{\mathrm{d}t}(n\boldsymbol{v}_h) = -g\nabla z - n\boldsymbol{v}_h\left(\nabla \cdot \boldsymbol{v}_h + \frac{\partial \omega}{\partial p}\right) - n\boldsymbol{f} \times \boldsymbol{v}_h. \tag{A4}$$

For simplicity, we show here derivation of the formulas in local Cartesian coordinates, even though spherical coordinates are used.

We assume that $n = \tilde{n}(p)$, which matches the similar assumption $\rho = \tilde{\rho}(z)$ applicable in $z$-coordinates, though the meaning is not identical. In the hydrostatic approximation, it holds $n \equiv 1$, so the assumption is trivially fulfilled.

With the assumption, for example the first equation from Eq. (A4) has the form

$$\frac{\partial}{\partial t}u + \frac{\partial}{\partial x}u^2 + \frac{\partial}{\partial y}(uv) + \frac{1}{\tilde{n}}\frac{\partial}{\partial p}(\tilde{n}u\omega) = -\frac{g}{\tilde{n}}\frac{\partial z}{\partial x} + fv. \tag{A5}$$

Finally, we divide the velocity components into the mean flow and gravity wave perturbation, $u = \langle u \rangle + u', v = \langle v \rangle + v'$ and

$\omega = \omega'$, and average the resulting equation over a rectangular area $A$. If we further assume the averaging fulfills

$$\int_A (\cdot)' \approx 0, \quad \frac{\partial}{\partial p}\int_A (\cdot)'\langle\cdot\rangle \ll \frac{\partial}{\partial p}\int_A (\cdot)'(\cdot)', \quad \int_{\partial' A} (\cdot)'\langle\cdot\rangle \approx 0, \tag{A6}$$

where $\partial' A$ denotes a face of the area $A$ (defined by coordinates $x_1$, $x_2$, $y_1$ and $y_2$), we get

$$\frac{\partial}{\partial t}\frac{1}{A}\int_A \langle u \rangle + \frac{1}{A}\left[\int_{\partial' A} \langle u \rangle^2 + (u')^2 \mathrm{d}y\right]_{x_1}^{x_2} + \frac{1}{A}\left[\int_{\partial' A} \langle u \rangle\langle v \rangle + u'v' \mathrm{d}x\right]_{y_1}^{y_2} + \frac{1}{\tilde{n}}\frac{\partial}{\partial p}\int_A \tilde{n}u'\omega'\mathrm{d}A = \frac{1}{A}\int_A fv_a \tag{A7}$$

with $v_a$ being the meridional ageostrophic wind component. Using this equation we can write the final formulas for the vertical

flux of zonal momentum and for three components of the zonal drag caused by gravity waves:

$$MF_{zx} = \frac{1}{A}\int_A \tilde{n}u'\omega'\mathrm{d}A, \tag{A8a}$$

$$GWD_{xx} = -\frac{1}{A}\left[\int_{\partial' A} (u')^2 \mathrm{d}y\right]_{x_1}^{x_2}, \tag{A8b}$$

$$GWD_{yx} = -\frac{1}{A}\left[\int_{\partial' A} u'v' \mathrm{d}x\right]_{y_1}^{y_2}, \tag{A8c}$$

$$GWD_{zx} = -\frac{1}{\tilde{n}}\frac{\partial}{\partial p}MF_{zx}. \tag{A8d}$$

Analogously, the formulas for meridional components are

$$MF_{zy} = \frac{1}{A}\int_A \tilde{n}v'\omega'\mathrm{d}A, \tag{A9a}$$

$$GWD_{xy} = -\frac{1}{A}\left[\int_{\partial' A} u'v' \mathrm{d}y\right]_{x_1}^{x_2}, \tag{A9b}$$

$$GWD_{yy} = -\frac{1}{A}\left[\int_{\partial' A} (v')^2 \mathrm{d}x\right]_{y_1}^{y_2}, \tag{A9c}$$

$$420 \quad GWD_{zy} = -\frac{1}{\tilde{n}} \frac{\partial}{\partial p} MF_{zy}. \tag{A9d}$$

In spherical coordinates, the formulas contain also terms emerging from derivatives of the geometry factors, analogously to equations in Procházková et al. (2023).

## A2  Fluxes and drag in zonal average

The derivation of the zonal averaged gravity wave momentum fluxes and drag is analogous to the previous derivation with the
difference of averaging Eq. (A4) over the parallels $l$ instead of the area $A$. Instead of the assumptions in Eq. (A6), we assume

$$\int_l (\cdot)' \approx 0, \quad \frac{\partial}{\partial p} \int_l (\cdot)'\langle \cdot \rangle \ll \frac{\partial}{\partial p} \int_l (\cdot)'(\cdot)', \quad \frac{\partial}{\partial \varphi} \frac{1}{l} \int_l (\cdot)'\langle \cdot \rangle \ll \frac{\partial}{\partial \varphi} \frac{1}{l} \int_l (\cdot)'(\cdot)', \tag{A10}$$

where $\varphi$ is the latitude.

Using the fact that the parallels are cyclic, we get by Eq. (A5)

$$\frac{\partial}{\partial t} \frac{1}{l} \int_l (\langle u \rangle + u') + \frac{1}{l} \int_l \frac{\partial}{\partial y} ((\langle u \rangle + u')(\langle v \rangle + v')) + \frac{1}{\tilde{n}} \frac{\partial}{\partial p} \frac{1}{l} \int_l (\tilde{n}(\langle u \rangle + u')\omega') = \frac{1}{l} \int_l fv_a. \tag{A11}$$

With the mentioned assumptions, it holds

$$\frac{1}{l} \int_l \frac{\partial}{\partial y} ((\langle u \rangle + u')(\langle v \rangle + v')) \mathrm{d}l = \frac{1}{2\pi R \cos\varphi} \int_0^{2\pi} \frac{1}{R} \frac{\partial}{\partial \varphi} ((\langle u \rangle + u')(\langle v \rangle + v')) R\cos\varphi \mathrm{d}\lambda$$

$$= \frac{1}{2\pi R} \frac{\partial}{\partial \varphi} \int_0^{2\pi} ((\langle u \rangle + u')(\langle v \rangle + v')) \mathrm{d}\lambda = \frac{\partial}{\partial \varphi} \frac{1}{2\pi R \cos\varphi} \frac{1}{R} \int_0^{2\pi} ((\langle u \rangle + u')(\langle v \rangle + v')) R\cos\varphi \mathrm{d}\lambda$$

$$= \frac{1}{R} \frac{\partial}{\partial \varphi} \frac{1}{l} \int_l ((\langle u \rangle + u')(\langle v \rangle + v')) \mathrm{d}l \approx \frac{1}{R} \frac{\partial}{\partial \varphi} \frac{1}{l} \int_l (\langle u \rangle\langle v \rangle + u'v') \mathrm{d}l. \tag{A12}$$

Here, $\lambda$ is the longitude, $R$ is the radial distance and the length of the parallel line $l$ equals $2\pi R \cos\varphi$. For the vertical term, we
have (assuming also relatively small changes of $\tilde{n}$ with longitude)

$$\frac{1}{\tilde{n}} \frac{\partial}{\partial p} \frac{\tilde{n}}{l} \int_l ((\langle u \rangle + u')\omega') \mathrm{d}l \approx \frac{1}{\tilde{n}} \frac{\partial}{\partial p} \frac{\tilde{n}}{l} \int_l u'\omega' \mathrm{d}l \tag{A13}$$

Therefore,

$$MF_{yx} = \frac{\tilde{n}}{l} \int_l u'v' \mathrm{d}l, \tag{A14a}$$

$$MF_{zx} = \frac{\tilde{n}}{l} \int_l u'\omega' \mathrm{d}l, \tag{A14b}$$

$$GWD_{xx} = 0, \tag{A14c}$$

$$GWD_{yx} = \frac{1}{R} \frac{\partial}{\partial \varphi} \frac{1}{l} \int_l u'v' \mathrm{d}l, \tag{A14d}$$

$$GWD_{zx} = \frac{1}{\tilde{n}} \frac{\partial}{\partial p} \frac{\tilde{n}}{l} \int_l u'\omega' \mathrm{d}l. \tag{A14e}$$

The same computations can be done for the meridional equation, resulting in formulas

$$MF_{yy} = \frac{\tilde{n}}{l} \int_l v'^2 \mathrm{d}l, \tag{A15a}$$

$$MF_{zy} = \frac{\tilde{n}}{l} \int_l v'\omega' \mathrm{d}l, \tag{A15b}$$

$$GWD_{xy} = 0, \tag{A15c}$$

$$GWD_{yy} = \frac{1}{R} \frac{\partial}{\partial \varphi} \frac{1}{l} \int_l v'^2 \mathrm{d}l, \tag{A15d}$$

$$GWD_{zy} = \frac{1}{\tilde{n}} \frac{\partial}{\partial p} \frac{\tilde{n}}{l} \int_l v'\omega' \mathrm{d}l. \tag{A15e}$$

## Appendix B: Additional figures

Figure B1 shows the average zonal wind on the subdomains.

Vertical profiles of GWD for their peak season evaluated using different filtering methods are displayed in Fig. B2. The triangular method provides very similar results to the rhomboidal method for equivalent cut off, with the rhomboidal method giving slightly lower amplitudes, which aligns with the comparison of the filtered fields in Fig. 3.

Figure B3 shows the vertical profile of the horizontal drag component GWDyx. The meridional derivative of the zonal flux of vertical momentum (GWDyx) has much lower values compared to the GWDzx component and it differs also qualitatively. The drag component is negative only in the lower stratosphere and becomes positive above 60 hPa. Mostly, the effects are however very low and the values are close to zero. A more significant GWDyx component can be observed only for the northern part of the Southern Andes, where the wind highly increases with height, causing also higher GWDzx component.

Figure B4 depicts the components of the zonal drag for the Southern Hemisphere subdomains, similarly to Fig. 10.

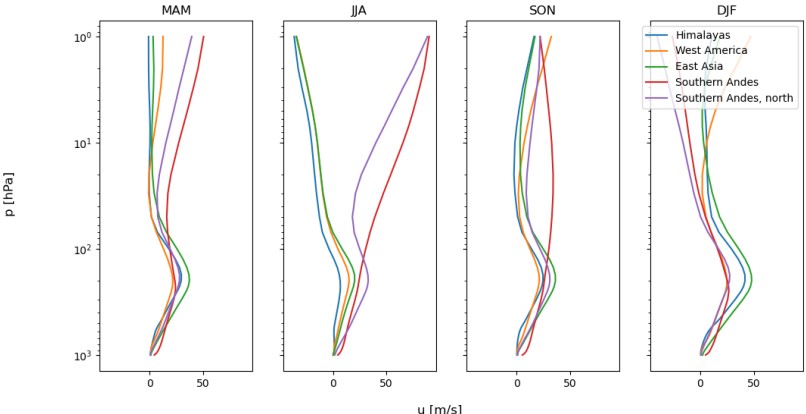

**Figure B1.** Zonal wind averaged on the subdomains.

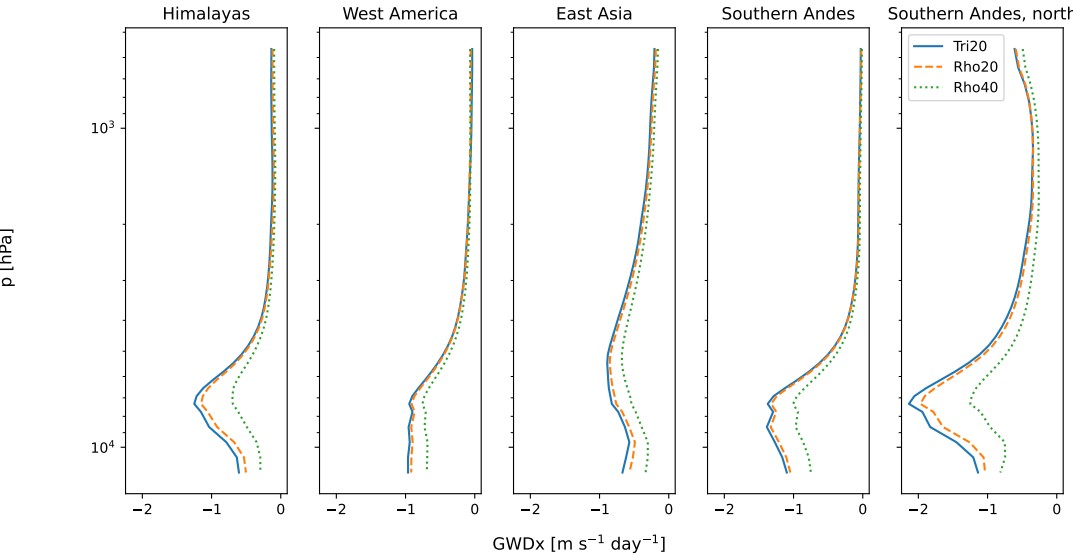

**Figure B2.** Vertical profiles of zonal MWD for peak seasons of the subdomains, evaluated after using different filtering method.

*Author contributions.* ZP: analysis of the raw data, methodology. RZ: MLR analysis. PŠ: conceptualization. All authors contributed to writing.

*Competing interests.* The authors declare that they have no conflict of interest.

| Phenomenon | Indices | Provider |
|---|---|---|
| NAO | Monthly NAO index | Climate Prediction Center, NOAA |
| ENSO | Multivariate ENSO Index Version 2 (MEI.v2) | Physical Science Laboratory, NOAA |
| QBO | Monthly mean zonal wind at 10, 30, 50 and 100 hPa at Singapore | Institute of Meteorology, Freie Universität Berlin |
| Solar cycle | Monthly mean total sunspot number | Royal Observatory of Belgium |

**Table B1.** Overview of used indices for MLR.

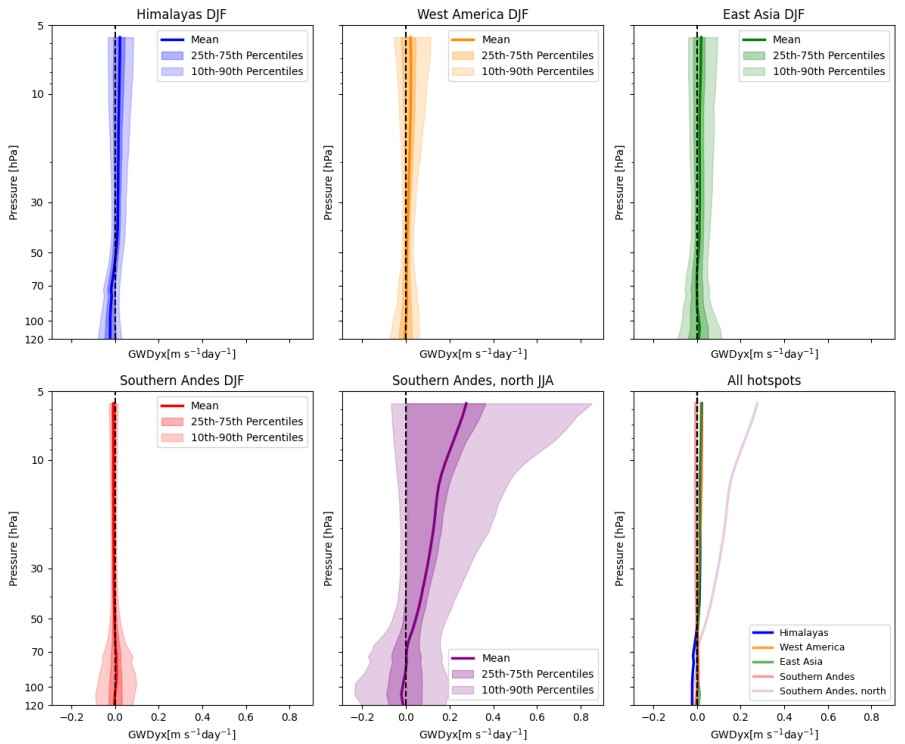

**Figure B3.** Climatological vertical distribution of GWDyx above the hotspots for their respective winters, obtained by Rho20 method.

*Acknowledgements.* This work was supported by the JUNIOR STAR project "Unravelling climate impacts of atmospheric internal gravity waves" under 23-04921M, using the resources of the Deutsches Klimarechenzentrum (DKRZ) granted by its Scientific Steering Committee (WLA) under project ID bm1233. PŠ was also partly supported by the Charles University Research Centre program No. UNCE/24/SCI/005.

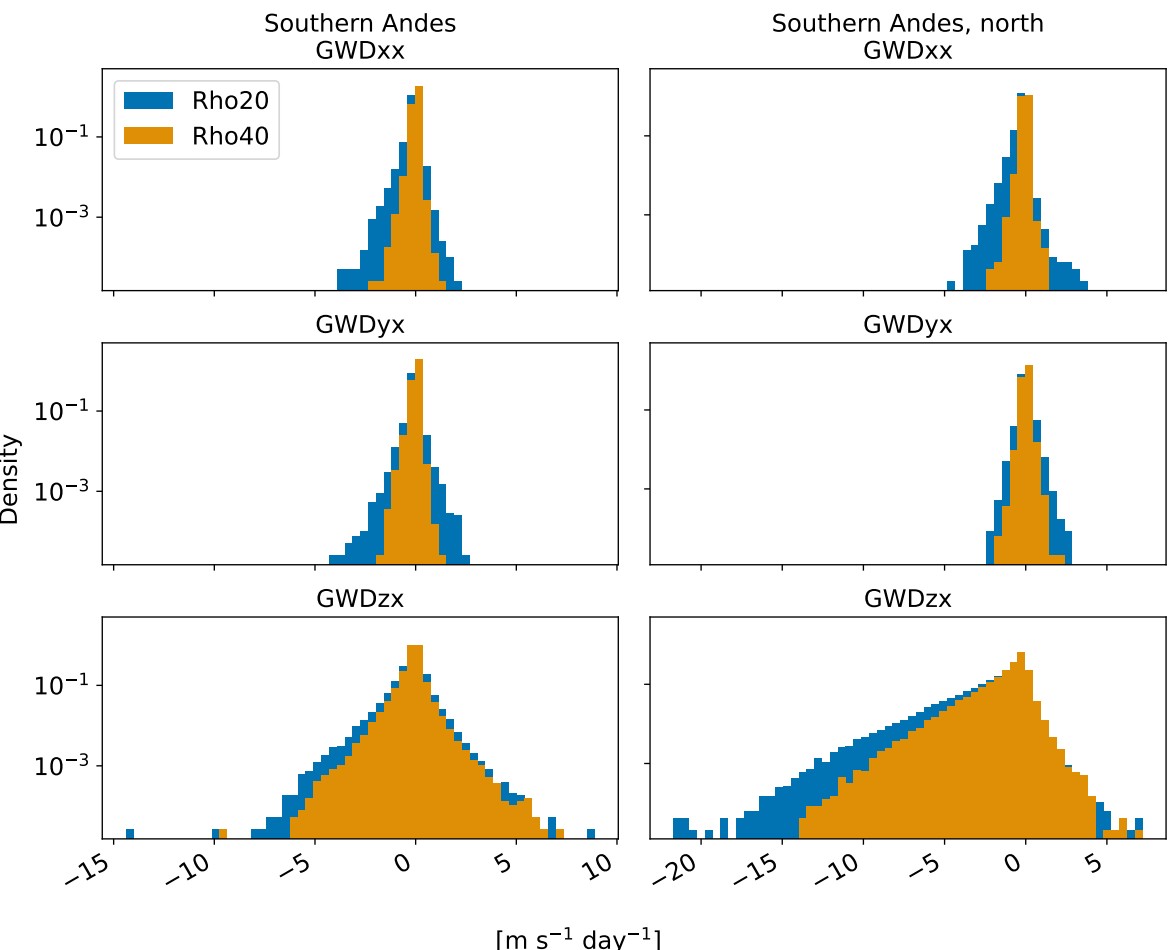

**Figure B4.** Histogram of GWDx for JJA on the Southern Hemisphere subdomains. The density is on a logarithmic axis.

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
