# Peer review of "Climatology, long-term variability and trend of resolved gravity wave drag in the stratosphere revealed by ERA 5"

_EGUsphere, 2025_

## Referee Comment (RC1)

**Review of EGUsphere-2025-939 "Climatology, long-term variability and trend of resolved gravity wave drag in the stratosphere revealed by ERA 5" by Z. Procházková , R. Zajíček, and P. Šácha**

**Overview**

The authors aim at giving a comprehensive overview of the climatology of gravity wave (GW) induced drag in the stratosphere. For this, 44 years of ERA 5 reanalysis data are investigated using a methodology first presented in Kruse et al. [2022] but adapted to weaker assumptions increasing the validity of this approach. Furthermore, they use a scale separation approach that has not yet been applied to the analysis of GWs in model data.

Using this toolbox, the authors present seasonal climatologies for zonal and meridional GW drag in the stratosphere as well as detailed climatologies of the usual GW hotspots, such as the southern Andes and Himalaya, giving insight into the variability and vertical structure of GW drag. Finally, the trend of GW drag throughout the investigated period is shown and the influence of other dynamical processes, such as the NAO, ENSO, solar activity, and the QBO, on GW drag is quantified.

In general, the paper is very well written and the scientific statements are well supported by the methodology and figures. At times, the text might need some smaller revision and some figures could be improved in visual presentation. Since a climatology of GW drag and a trend analysis is unprecedented, my verdict of the article is that it is well worth publishing in Weather and Climate after a minor revision.

**General comments**

1. Your analysis takes into account regional differences by your separation into "GW hotspots". Since your analyzed period is quite long, it would be awesome if you could also investigate the differences in your GW hotspots of the northern hemisphere in terms of variability between SSW and non-SSW years. In particular, I would be interested in the Himalaya region, as it is hypothesized that strong GW activity in this region might precondition the vortex before an SSW. A climatology of the SSW vs non-SSW periods in this region might shed some light into the role of the Himalaya. Did you have a look into this? For me, it's not necessary to include it in this publication if it's more work than I imagine, however, it would be an interesting point for a short follow up study.

**Specific comments**

- l16: '...phenomenon in the terrestrial atmosphere...' GWs are not limited to the terrestrial atmosphere but are also present on other planets with an atmosphere (e.g. on Mars)

- l35f: '..., which cannot be to date derived from global-scale observations.' rephrase this sentence. I think you mean that there are nor current observations that allow for this derivation. However, it could be understood as that the methodologies for extracting the GW parameters do not yet exist, which would be wrong [cf. Hindley et al., 2020, Lear et al., 2024, Rhode et al., 2024].

- l63: When mentioning ERA 5.1, it would be helpful to explain why you are using it. I.e., half a sentence on what ERA 5.1 improves upon compared to ERA 5.

- l111-113: 'A perfect match between vertical velocity and horizontal velocity perturbations cannot be expected, as polarization relations indicate that gravity waves with shorter horizontal wavelengths tend to project onto horizontal wind perturbations and vice-versa.' This sentence is not clear to me and should be rephrased. Do you mean that shorter waves have higher ratio of vertical to horizontal wind amplitudes (which would be the case from the polarization relations) or is it about a direction change (as implied by the word 'projection')?

- 179 & Fig. 5: 'Additionally, we see the...' this is barely visible. The figure is definitely too small to see all the features well. Consider making it wider. You could gain some width be dropping the y ticks on the middle an right panels, as they are the same(?). Please also add a period axis as top axis for the plots, which would make figuring out the features much easier than counting the dotted lines from left to right. The red line is barely visible.
  This figure would probably benefit from showing only 2 columns but wider panels.

- Fig. 6: Consider adding a running mean and limit the y axis to Jan-Jan. West America is not visible at all, maybe the visibility could be improved by adding transparency to the high resolution data shown here and a solid-colored running mean (e.g. 7 days).

- Fig. 7 & 8: Why are you using transparency in panel f? Is this for color blindness? If so, make sure that Fig. 6 is also accessible for color blind people.

- 'A possible reason for this can be that the horizontal scales and background flow of the Himalaya hotspot favor sourcing of longer orographic GW modes that propagate mostly vertically.' This could be misunderstood as "GWs with longer horizontal wavelengths propagate mostly vertical" which would be false in general. I think, you mean something like: In the Himalaya region, the longer horizontal GWs seem to show faster vertical propagation. The reason for this, however remains unclear (do they also have longer vertical wavelengths? why is this only the case in the Himalaya region? are the winds different here than in other orographic GW regions?). Please rephrase in a way that states what you mean more clearly.

**Technical comments**

- l18: '...GWs exist **at** and...'

- l21: 'on a leading order' → *at leading order*

- l46, 48: consider adding Lear et al. [2024] to the listed citations.

- l66: drop 'the' in 'using the linear interpolation'

- l68: drop 'the' in 'above the 10 hPa'

- l71: Consider changing '. The details of the filtering are described in the following subsection.' to *as described in Sec. 2.1*

- l131: Consider dropping 'the years'.

- l160: Add a comma: 'In their work**,** the meridional...'

- l167: Maybe: 'The zonal mean meridional drag component has by more than a half smaller magnitude...' → *The magnitude of the zonal mean meridional drag component is by a factor of more than 2 smaller than the zonal component.*?

- l189f: 'For Southern Andes, based on the definition of the hotspot, two opposite yearly cycles can be derived.' is not too clear. Maybe something like: *For the Southern Andes regions (as shown in Fig. 1), two opposing yearly cycles can be derived.*'

- l193: 'overlaying' → *overlapping*

- l200: 'maximum **is** not'

- l217: 'which is **asymmetrical** around'

- l218: 'The probability distributions in Fig. 9 for two cut-off ranges also reveal' is not clear, please rephrase. E.g.: *he probability distributions for different scale-separation cut offs in Fig. 9 also reveal.*

- l268: Maybe: 'The last column...' → *The **rightmost** column...*?

- l292: '...ozone **depletion** and recovery...'

- l322 & 327: *drag* instead of 'drags'

- l339: no brackets around citation

- l353: '...GW field**s**...'

- l354: 'dedicated **to** this effort'

- l358: Maybe *globally observe local GW parameters and induced GW drag*

**References**

N. P. Hindley, C. J. Wright, L. Hoffmann, T. Moffat-Griffin, and N. J. Mitchell. An 18-year climatology of directional stratospheric gravity wave momentum flux from 3-D satellite observations. *Geophys. Res. Lett.*, 47(22):e2020GL089557, NOV 28 2020. ISSN 0094-8276. doi: 10.1029/2020GL089557.

C. G. Kruse, M. J. Alexander, L. Hoffmann, A. van Niekerk, I. Polichtchouk, J. Bacmeister, L. Holt, R. Plougonven, P. Sacha, C. Wright, K. Sato, R. Shibuya, S. Gisinger, M. Ern, C. Meyer, , and O. Stein. Observed and modeled mountain waves from the surface to the mesosphere near the Drake Passage. *J. Atmos. Sci.*, pages 909–932, 2022. doi: 10.1175/JAS-D-21-0252.1.

E. J. Lear, C. J. Wright, N. P. Hindley, I. Polichtchouk, and L. Hoffmann. Comparing gravity waves in a kilometer-scale run of the ifs to airs satellite observations and era5. *Journal of Geophysical Research: Atmospheres*, 129(11):e2023JD040097, 2024. doi: https://doi.org/10.1029/2023JD040097. URL `https://agupubs.onlinelibrary.wiley.com/doi/abs/10.1029/2023JD040097`. e2023JD040097 2023JD040097.

S. Rhode, P. Preusse, J. Ungermann, I. Polichtchouk, K. Sato, S. Watanabe, M. Ern, K. Nogai, B.-M. Sinnhuber, and M. Riese. Global scale gravity wave analysis methodology for the ESA Earth Explorer 11 candidate CAIRT. *Atmos. Meas. Tech.*, 17:5785–5819, 2024. doi: 10.5194/amt-17-5785-2024. URL `https://amt.copernicus.org/articles/17/5785/2024/`.

---

## Author Comment (AC1)

First, we would like to thank both the referees and the editor Prof. Dr. Jürg Schmidli for facilitating an open minded and constructive review process. The thoughtful comments and well-minded suggestions of both reviewers helped us to improve the readability and clarity of the text and graphics in the revised version.

In particular, we would like to thank the Reviewer#1 for numerous helpful comments and suggestions. Below, we answer the general and specific comments (reviewer comments in blue, answers in black). All technical comments were modified accordingly.

**General comments**

1. Your analysis takes into account regional differences by your separation into "GW hotspots". Since your analyzed period is quite long, it would be awesome if you could also investigate the differences in your GW hotspots of the northern hemisphere in terms of variability between SSW and non-SSW years. In particular, I would be interested in the Himalaya region, as it is hypothesized that strong GW activity in this region might precondition the vortex before an SSW. A climatology of the SSW vs non-SSW periods in this region might shed some light into the role of the Himalaya. Did you have a look into this? For me, it's not necessary to include it in this publication if it's more work than I imagine, however, it would be an interesting point for a short follow up study.

Thank you for the comment. We share your interest in the hypothesis about a possible role of Himalayas for the polar vortex preconditioning. The drag time series computed for this manuscript can certainly be used for this goal, however, we prefer not to include any composite results in the current manuscript nor to disclose any similar results at this point, as we are currently preparing several manuscripts on the dynamical effects of the resolved GW drag, with one study targeting particularly the Himalayan hotspot. That said, we decided to provide the resulting GWD time-series subject to the potential publication of our manuscript (in daily averages and on reduced vertical grid to obtain reasonable size of the data; 10.5281/zenodo.15473685) and we hope that this will be useful for the community for a quick look also on the potential dynamical links like this.

**Specific comments**

• l16: '...phenomenon in the terrestrial atmosphere...' GWs are not limited to the terrestrial atmosphere but are also present on other planets with an atmosphere (e.g. on Mars)

Thank you for the correction, the word "terrestrial" was removed.

• l35f: '..., which cannot be to date derived from global-scale observations.' rephrase this sentence. I think you mean that there are nor current observations that allow for this derivation. However, it could be understood as that the methodologies for extracting the GW parameters do not yet exist, which would be wrong [cf. Hindley et al., 2020, Lear et al., 2024, Rhode et al., 2024].

Thanks for pointing this out. Reformulated to "…which cannot currently be derived due to the lack of suitable global-scale observational data."

• l63: When mentioning ERA 5.1, it would be helpful to explain why you are using it. I.e., half a sentence on what ERA 5.1 improves upon compared to ERA 5.

Text modified to: "We base our study on hourly ERA 5 data on model levels with the horizontal resolution corresponding to about 31 km (0.28125 degrees) for the period 1979 – 2023 (Hersbach et al., 2017), combined with ERA 5.1 data, which correct stratospheric temperature bias present in ERA 5 for years 2000 – 2006 (Simmons et al., 2020)."

• l111-113: 'A perfect match between vertical velocity and horizontal velocity perturbations cannot be expected, as polarization relations indicate that gravity waves with shorter horizontal wavelengths tend to project onto horizontal wind perturbations and vice-versa.' This sentence is not clear to me and should be rephrased. Do you mean that shorter waves have higher ratio of vertical to horizontal wind amplitudes (which would be the case from the polarization relations) or is it about a direction change (as implied by the word 'projection')?

Thanks for the comment and suggestion. We indeed mean the ratio between the amplitudes. We reformulated the text to "A perfect match between vertical velocity and horizontal velocity perturbations cannot be expected, as polarization relations indicate that gravity waves with shorter horizontal wavelengths tend to have higher ratio of vertical to horizontal wind amplitudes and vice-versa." Further, we added a reference to the polarisation relation in Fritts and Alexander, 2003.

• 179 & Fig. 5: 'Additionally, we see the...' this is barely visible. The figure is definitely too small to see all the features well. Consider making it wider. You could gain some width be dropping the y ticks on the middle an right panels, as they are the same(?). Please also add a period axis as top axis for the plots, which would make figuring out the features much easier than counting the dotted lines from left to right. The red line is barely visible. This figure would probably benefit from showing only 2 columns but wider panels.

Thanks for pointing this out. The figure was re-shaped, the red line was changed from dotted to dashed and we added labels with the period to the dotted line.

• Fig. 6: Consider adding a running mean and limit the y axis to Jan-Jan. West America is not visible at all, maybe the visibility could be improved by adding transparency to the high resolution data shown here and a solid-colored running mean (e.g. 7 days).

Thank you for the tip – the plot was adjusted to show the transparent lines and the running mean and the x axis was limited to Jan-Jan.

• Fig. 7 & 8: Why are you using transparency in panel f? Is this for color blindness? If so, make sure that Fig. 6 is also accessible for color blind people.

Thanks. We used the transparency in Fig. 6 as you suggested in the previous comment. Additionally, we checked the figures in a colorblind simulator to be sure that the selected colors are accessible for people with the common types of color blindness.

• 'A possible reason for this can be that the horizontal scales and background flow of the Himalaya hotspot favor sourcing of longer orographic GW modes that propagate mostly vertically.' This could be misunderstood as "GWs with longer horizontal wavelengths

propagate mostly vertical" which would be false in general. I think, you mean something like: In the Himalaya region, the longer horizontal GWs seem to show faster vertical propagation. The reason for this, however remains unclear (do they also have longer vertical wavelengths? why is this only the case in the Himalaya region? are the winds different here than in other orographic GW regions?). Please rephrase in a way that states what you mean more clearly.

Reformulated to "A possible reason for this can be that the horizontal scales and geometry of the Himalaya orography together with its orientation with respect to the predominantly zonal background flow favor sourcing of longer orographic GW modes that propagate vertically more efficiently compared to longer GWs for other hotspots."

Technical comments

• l18: '...GWs exist at and...'

Modified.

• l21: 'on a leading order' → at leading order

Modified.

• l46, 48: consider adding Lear et al. [2024] to the listed citations.

Added.

• l66: drop 'the' in 'using the linear interpolation'

Removed.

• l68: drop 'the' in 'above the 10 hPa'

Removed.

• l71: Consider changing '. The details of the filtering are described in the following subsection.' to as described in Sec. 2.1

Modified to "horizontal velocity perturbations due to GWs are separated, as described in Sec. 2.1".

• l131: Consider dropping 'the years'.

Removed.

• l160: Add a comma: 'In their work, the meridional...'

Added.

• l167: Maybe: 'The zonal mean meridional drag component has by more than a half smaller magnitude...' → The magnitude of the zonal mean meridional drag component is by a factor of more than 2 smaller than the zonal component.?

Changed to "is by a factor of more than two smaller".

• l189f: 'For Southern Andes, based on the definition of the hotspot, two opposite yearly cycles can be derived.' is not too clear. Maybe something like: For the Southern Andes regions (as shown in Fig. 1), two opposing yearly cycles can be derived.'

Modified.

• l193: 'overlaying' → overlapping

Modified.

• l200: 'maximum is not'

Corrected.

• l217: 'which is asymmetrical around'

Corrected.

• l218: 'The probability distributions in Fig. 9 for two cut-off ranges also reveal' is not clear, please rephrase. E.g.: he probability distributions for different scale-separation cut offs in Fig. 9 also reveal.

Changed to "The probability distributions for different scale-separation cut-offs in Fig. 9 also reveal"

• l268: Maybe: 'The last column...' → The rightmost column...?

Modified.

• l292: '...ozone depletion and recovery...'

Corrected.

• l322 & 327: drag instead of 'drags'

Modified.

• l339: no brackets around citation

Changed to a citation with e.g. in the bracket.

• l353: '...GW fields...'

Corrected.

• l354: 'dedicated to this effort'

Corrected.

• l358: Maybe globally observe local GW parameters and induced GW drag

Modified.

---

## Author Comment (AC2)

First, we would like to thank both the referees and the editor Prof. Dr. Jürg Schmidli for facilitating an open minded and constructive review process. The thoughtful comments and well-minded suggestions of both reviewers helped us to improve the readability and clarity of the text and graphics in the revised version.

In particular we would like to thank the Reviewer#2 for his comments, suggestions and thoughtful questions and points raised. We carefully considered all the comments and suggestions. Below, you will find the original reviewer's comments (in blue) and our answers (in black).

Major comments:

1. However, it also appears to me that the role of the mean flow in shaping the drag and its seasonal cycle could be worked out more nicely. This mostly concerns the seasonality in the southern hemisphere over the Andes, where the southernmost region is not exhibiting most intensive fluxes in winter, but in summer. If I understand them correctly, the authors argue that this is because the mean zonal wind in this region does not reverse its vertical derivative in the lower stratosphere. They claim that this is in line with linear wave dynamics. This might well be true, but I think it should be demonstrated. This could be done by single-column calculations of the drag given the diagnosed winds, with a simple wave emitted in the troposphere. Analytical arguments would be very welcome as well.

We would like to thank the referee for pointing out that the reasoning behind the importance of the zonal wind curvature above the UTLS jet maximum and of the valve (zero wind) layer has not been clear enough. For illustrating the connection of GW breaking with the zonal background winds we repeat here the saturation flux argument developed originally by Lindzen (1981), which estimates a maximal momentum flux that can be propagated vertically without breaking for a given zonal wind profile:

One can derive (see also e.g. Fritts, 1984) that the following equality holds for convectively unstable regions, $u' + \overline{u} = c$, where $u'$ is the wind perturbation, $\overline{u}$ is the mean wind and $c$ is the phase velocity. Taking the limiting case for the saturated monochromatic waves $|u'_s| = |\overline{u} - c|$, considering the polarisation relations and the equality for the vertical wavenumber $m = N/(\overline{u} - c)$ in the WKB settings, with the Brunt-Vaisala frequency $N$, we can write

$$w_s' = -\frac{k}{m}u_s' = \frac{k}{N}(\overline{u} - c)^2.$$

Therefore, the maximal vertical flux component can be written as

$$\overline{\rho}\,\overline{u_s'w_s'} = \frac{1}{2}\overline{\rho}\,u_s'w_s' = -\frac{1}{2}\overline{\rho}\,\frac{k}{N}(\overline{u} - c)^3.$$

In some form, this criterion is central to a majority of GW parameterization schemes for identifying regions of GW dissipation in the free atmosphere. It makes explicit that in the situation of the increasing wind speed in the vertical, the wind field counteracts

the exponential decrease of density with height and the wave can propagate vertically without reaching the convective instability threshold.

Above the center of the UTLS jet, the wind is no longer increasing with height and the saturation flux decreases sharply indicating a potential GW dissipation region. Moreover, for orographic GWs, the region above the UTLS jet is important also because of the occurrence of a region of near zero winds (termed also the valve layer or a neck region), which serve as a critical level for OGW propagation.

It has to be noted that the curvature of the wind above the UTLS jet may be too strong for the WKB approximation to hold and the existence of breaking cannot be assessed rigorously by a simple nonlocal argument. From this perspective, we refer to our results showing a clear GWD maximum above the UTLS jet as reassuring, documenting the general validity of the simple saturation criterion.

Changes in text:

- We added reference to the saturation hypothesis after mentioning it in the context of the zonal mean profiles (L146) and we expanded the explanation:

  "It is a direct consequence of the saturation criterion employed in the parameterizations (Lindzen, 1981; Nappo, 2012) that requires momentum flux convergence in the region of negative wind shear above the jet center, where the increase of the mean wind does not balance the decrease of the density anymore."

- To the explanation of the missing winter minimum for the southernmost Southern Andes subdomain (L202), we added the text: "Therefore, during the Southern Hemisphere winter, the vertical profile of the zonal wind above the subdomain does not suggest any regions of potential instability or critical level filtering according to the saturation hypothesis and the resolved wave field behavior confirms this."
- To increase clarity, corresponding section in the discussion (L311) reformulated to:

  "The zonal mean drag climatology in the stratosphere exhibits a vertical distribution consistent with the saturation hypothesis, as predicted in simplified models, featuring pronounced GWD maxima above the center of the UTLS jet across all seasons in both hemispheres."

Lindzen, R. S. (1981), Turbulence and stress owing to gravity wave and tidal breakdown, *J. Geophys. Res.*, 86(C10), 9707–9714, doi:10.1029/JC086iC10p09707.

Fritts, D. C. (1984). Gravity wave saturation in the middle atmosphere: A review of theory and observations. *Reviews of Geophysics*, *22*(3), 275-308.

2. The authors decide for a rhomboidal instead of a triangular horizontal spectral filter in order to extract the gravity-wave signal from the horizontal wind. It is not quite clear to me whether this is the more appropriate approach. After all it should be the total wave number that decides, not the meridional wave number. Zonally symmetric gravity waves are not excluded by definition. Whatever, the authors state that the triangular filter gives the same results. It would be good to demonstrate this with a single figure.

Thank you for commenting on this methodological aspect. The advantage of the rhomboidal truncation is the possibility to potentially adjust the filter according to one's needs and not-allowing modes that are longer in both zonal and meridional directions (see Fig. 2 in the manuscript). In our understanding, the triangular truncation is not something clearly related to GW.  In this light, we see the rhomboidal truncation as superficial to the triangular truncation for GW separation. But, as we already stated in the manuscript, in the average sense the two methods give almost identical results and we document this by adding a figure with the comparison of the GW drag by the methods in the appendix as Fig. B2, which is also shown below.

Also, we added a description of the figure:

"Vertical profiles of GWD for their peak season evaluated using different filtering methods are displayed in Fig. B2. The triangular method provides very similar results to the rhomboidal method for equivalent cut off, with the rhomboidal method giving slightly lower amplitudes, which aligns with the comparison of the filtered fields in Fig. 3."

and a reference to the figure to Section 3.2:

"The vertical profiles do not deviate significantly from profiles that would be obtained by the filtering with triangular truncation (see Fig. B2)."

[Figure]

Minor comments:

l. 70 and Fig. 3: Would it not be more consistent to also filter the vertical wind? True, linear theory tells us that to leading order all vertical wind is gravity waves, but there is large-scale balanced dynamics for vertical flow (omega equation yielding the balanced ageostrophic flow) so that some large-scale vertical wind cannot really be attributed to gravity waves.

Thank you for mentioning that. We agree that filtering of the vertical wind would be an option. However, due to the large uncertainty in the filtering methods (e.g., the selection of the cutoff), the filtering would add this uncertainty to the field, and the low-passed vertical wind appears to be generally very small (see e.g., Fig. 2 in Sun at al, 2023). In any case, this should be definitely discussed more in the paper and we modified the text in the methodology (L72 in the revised manuscript). We changed

"(the vertical velocity field is assumed to be dominated by GWs in the stratosphere and hence no filtering is needed)."

-> "We do not apply any filtering of the vertical velocity field as the theory and existing literature (Sun et al., 2023) suggest the dominance of the GW perturbations to the leading order and a filtering procedure might possibly introduce some artifacts to the resulting fields."

Sun, Y. Q., Hassanzadeh, P., Alexander, M. J., & Kruse, C. G. (2023). Quantifying 3D gravity wave drag in a library of tropical convection‐permitting simulations for data‐driven parameterizations. *Journal of Advances in Modeling Earth Systems*, *15*(5), e2022MS003585.

l. 77: I understand that the factor n was necessary in Prochazkova et al (2023) where WRF data had been analyzed, i.e. from a non-hydrostatic model. However, here IFS data are used, i.e. from a hydrostatic model where by definition n = 1. This should not be presented as an assumption but rather as a consequence of the model formulation. Perhaps one could even set n = 1 directly?

Thank you for the comment. The derivation of the equation is shown with the variable n for the generality. However, as mentioned in the methodology subsection, for most parts of the dataset, we use n=1 as the computation of n for upper levels significantly increases the computational costs. Regarding the 5 lower levels, the values of n actually range between 0.9 and 1.1 with the average at 1. These differences from the mean are present because although IFS is a hydrostatic model, the assimilation schemes in the ERA 5 reanalysis are slightly pushing it away from the hydrostatic equilibrium. Therefore, we prefer to leave the equations in a more general form with a variable n.

l. 167 – 168: Not quite sure whether the weakness of the zonal-mean meridional circulation is a good argument why weaker meridional drag is significant. To leading order, the time-mean residual circulation is determined by the zonal drag!

Thank you for identifying a possibly confusing formulation. The term zonal mean meridional circulation probably gave an impression that we imply some sort of nonlocal dynamical

mechanism. Sorry for this. The text was modified so that it does not suggest the interpretation of means from:

"However, for context, the zonal mean meridional circulation is many times slower than the zonal mean zonal winds."

to

"However, since the meridional wind is generally lower than the zonal wind, weaker meridional drag can have a relatively strong effect on the circulation."

l. 174: Even without oblique propagation, in classic simple single-column gravity-wave parameterizations horizontal fluxes and their convergence are allowed. They are not considered, for simplicity, but they are there. Oblique propagation will horizontally redistribute vertical and horizontal fluxes.

Thanks for pointing out the misleading formulation. Sentence

"Our methodology allows to compute regional drag estimates, which we do in the following for selected major extratropical hotspots, taking fully into account the effects of the oblique propagation"

 changed to

"Our methodology allows us to compute regional drag estimates, which we do in the following for selected major extratropical hotspots, evaluating also the effects of horizontal fluxes."

l. 179 – 180: A diurnal signal in the gravity-wave drag could also be a signature of coupling with solar tides.

Excellent comment, thanks for the remark. We added a sentence (L187 in the revised manuscript)

"This can be caused either directly by solar heating of the air and the Earth's surface or secondarily by coupling with the solar tides."

Fig. 5: Please mention in the caption that the spectra are for 70hPa.

Added.

Fig. 7 and 8: Replace 'winter' by 'peak season'?

Thanks, it is now corrected.

l. 231: Should it not be shorter (instead of longer) orographic waves that propagate mostly in the vertical direction?

Thanks for pointing out the possibly misleading formulation. Based of your comment and the comment of the other reviewer, we changed the formulation to:

"A possible reason for this can be that the horizontal scales and geometry of the Himalaya orography together with its orientation with respect to the predominantly zonal background flow favor sourcing of longer orographic GW modes that propagate vertically more efficiently compared to longer GWs for other hotspots."

---

## Author Response (AR2)

1.       Please add the equation for the upper limit of the vertical flux component in the 1st paragraph of section 3.1. Not all readers might have it ready in their mind, and so the manuscripts would increase in readability.

Based on the reviewer's comment, we added the equation with the following description to the first paragraph in section 3.1:

This follows from an estimate of the maximum momentum flux that can be propagated vertically without breaking for a given zonal wind profile. Assuming motions confined to the x–z plane, let $\bar{u}$ represent the vertical profile of the mean wind in the x-direction, k the horizontal wavenumber, c the phase velocity, $\bar{\rho}$ the mean density profile and N the Brunt-Väisälä frequency. Under these assumptions, the upper limit on the momentum flux can be expressed as follows  (Fritts 1984):

$$MF^{s}_{zx} = -\frac{1}{2}\bar{\rho}\,\frac{k}{N}\,(\bar{u} - c)^{3},$$

which highlights the dependence of the maximum momentum flux on the background wind profile.

2.       I am still not quite convinced about the authors response to my comment on l. 167 – 168. As long as not averaged zonally, meridional winds are as strong as the zonal winds in any eddies. In steady state the zonal mean meridional residual wind is not influenced by the meridional gravity-wave drag. I would recommend that the authors be careful about this, but would leave it to them whether they want to further adjust the text.

Thank you for pointing out the source of the confusion. To make the text more clear, we decided to remove the ambiguous sentences that are anyway not important for the overall conclusions of the paper - "The zonal mean meridional drag component is by a factor of more than two smaller than the zonal component. However, since the meridional wind is generally lower than the zonal wind, weaker meridional drag can have a relatively strong effect on the circulation."